# O3D: Offline Data-driven Discovery and Distillation for Sequential Decision-Making with Large Language Models

**Yuchen Xiao**\*, **Yanchao Sun**\*, **Mengda Xu, Udari Madhushani Sehwag, Jared Vann**
**Deepeka Garg, Sumitra Ganesh**
J.P. Morgan AI Research
{yuchen.xiao, yanchao.sun, mengda.xu}@jpmchase.com
{udari.madhushani.sehwag, jared.vann, deepeka.garg, sumitra.ganesh}@jpmorgan.com

## Abstract

Recent advancements in large language models (LLMs) have exhibited promising performance in solving sequential decision-making problems. By imitating few-shot examples provided in the prompts (i.e., in-context learning), an LLM agent can interact with an external environment and complete given tasks without additional training. However, such few-shot examples are often insufficient to generate high-quality solutions for complex and long-horizon tasks, while the limited context length cannot consume larger-scale demonstrations with long interaction horizons. To this end, we propose an offline learning framework that utilizes offline data at scale (e.g, logs of human interactions) to improve LLM-powered policies without finetuning. The proposed method *O3D (Offline Data-driven Discovery and Distillation)* automatically discovers reusable skills and distills generalizable knowledge across multiple tasks based on offline interaction data, advancing the capability of solving downstream tasks. Empirical results under two interactive decision-making benchmarks (ALFWorld and WebShop) verify that O3D can notably enhance the decision-making capabilities of LLMs through the offline discovery and distillation process, and consistently outperform baselines across various LLMs.

## 1 Introduction

Recent years have witnessed remarkable advancements in artificial intelligence (AI), particularly in the development of Large Language Models (LLMs). One of the standout features of LLMs is their in-context learning ability, where the LLM can perform tasks with only a few-shot examples provided in the prompts, making it possible to deploy LLMs to various applications seamlessly.

Although most existing research focuses on one-step text generation such as question answering, many real-world scenarios desire autonomous agents that can interact with external environments and make sequential decisions to complete given tasks. Some recent works successfully showcase the application of LLMs in sequential decision-making (Yao et al., 2023b; Shinn et al., 2023; Liu et al., 2023c; Yang et al., 2023), by either directly providing few-shot examples of acting and reasoning in the prompt (Yao et al., 2023b), or learning via trial and error during online interactions with the environment (Shinn et al., 2023).

However, existing methods still struggle to solve many complex domains with LLMs due to the intrinsic difficulties that arise from long-horizon interactive tasks. On the one hand, it is widely known that the task complexity increases exponentially with the interaction horizon (Sutton & Barto, 2018), such that a large amount of data or demonstrations can be desired for an agent to fully understand the environment dynamics, especially for heterogeneous and stochastic real-world tasks, where cross-task generalizability is important. On the other hand, the in-context learning ability is constrained by the limited context window

---

\*Equal Contribution

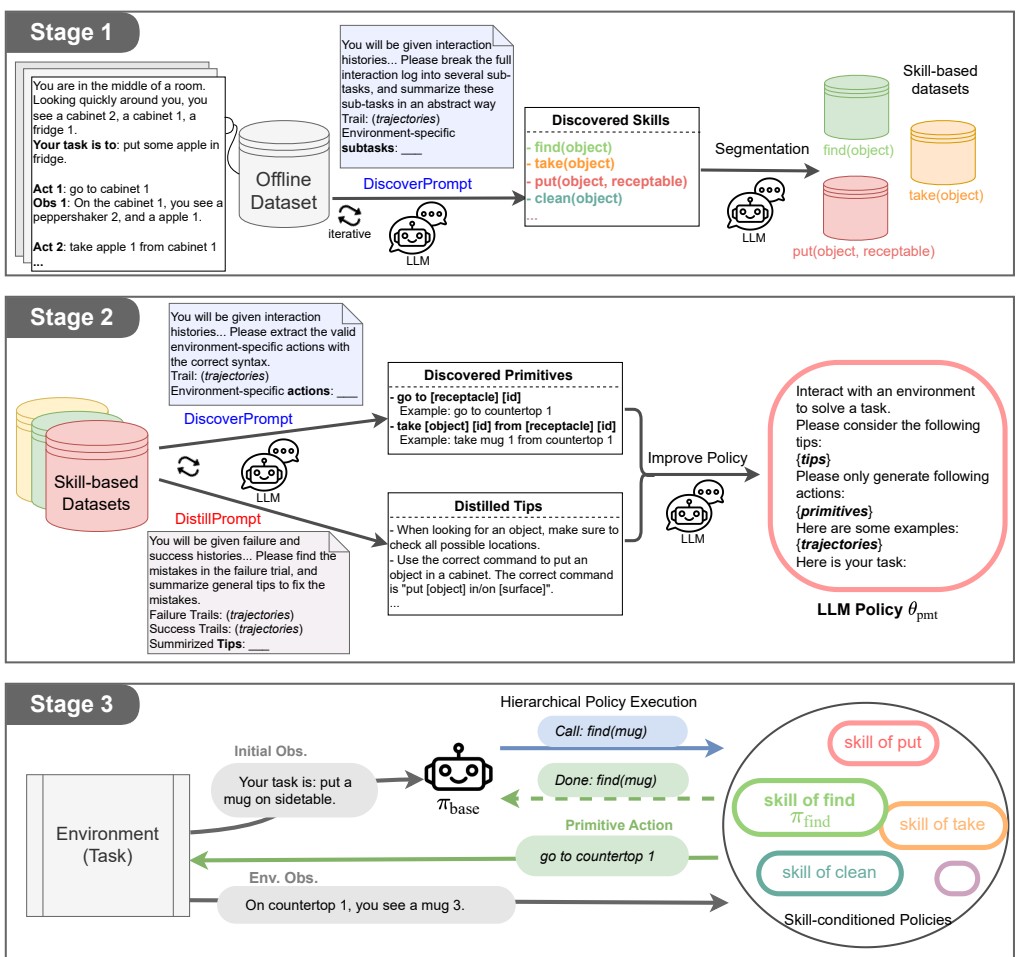

Figure 1: The proposed O3D framework. Stage 1: offline skill discovery and data segmentation. Stage 2: offline policy improvement with knowledge distillation. Stage 3: downstream interaction with hierarchical policy execution.

of an LLM. Even if many demonstrations exist, it is hard and inefficient to prompt LLMs with a large number of examples. Although finetuning is a solution, it can be much more expensive and less accessible for normal users.

In response to these challenges, this paper asks and aims to answer the follow question:

*Can we develop a data-driven learning framework for sequential decision-making with LLMs, which learns from large-scale offline data without the need of model training?*

In this paper, we define an offline learning framework to enable LLMs to discover and distill useful knowledge from interaction trajectories on multiple tasks. For LLM-powered agents/policies, we carefully design a generic learning paradigm called *Offline Data-driven Discovery and Distillation* (O3D), which can iterate over the offline dataset and keep improving the policy. Importantly, our method does not require a high-quality expert offline dataset, as it can benefit from both positive examples and negative examples of environment interactions, making the framework easier and cheaper to use in practice. As shown in Figure 1, O3D is composed of 3 stages: the first stage aims to discover reusable skills by segmenting the offline interaction trajectories; the second stage then conducts skill-conditioned policy improvement by distilling knowledge from offline data; the third stage constructs the interactive policy by calling the learned skills given diverse tasks. All stages are based on querying LLMs and iterating over existing datasets, to improve or utilize LLM-powered policies. As a result, O3D can learn better policies from offline dataset at scale without any model finetuning. Experiments in two commonly used domains (ALFWorld and Web-

Shop) show that our LLM agent augmented by offline knowledge has much better few-shot performance than prior methods on various tasks.

**Summary of contributions: (1)** We establish an offline in-context learning framework for LLM sequential decision-making agents, such that the LLM can learn from offline experience without finetuning. This offline learning paradigm allows more effective usage of past interactions (including both good and bad behaviors) generated by humans or other agents, alleviating the cost of online learning. **(2)** Different from prior work which prompts and solves different types of tasks independently, our algorithm leverages offline experience from multiple tasks. By letting LLMs automatically distill shared high-level knowledge, our algorithm achieves few-shot generalization to various types of tasks with a single set of prompts. **(3)** O3D is a versatile framework that can boost the performance of existing methods. Empirical results show that the knowledge distilled by O3D can be combined with existing in-context learning approaches to achieve superior performance as shown in Section 4. O3D can also facilitate code generation in various tasks, as detailed in Appendix D.

## 2   Related Work

**In-context learning of LLMs for sequential decision-making.** *(1) Generating text.* Yang et al. (2023) and Liu et al. (2023b) conduct extensive experiments to showcase the ability of LLMs to make sequential decisions in a range of challenging multi-step reasoning tasks. Yao et al. (2023b) propose ReAct, a method to combine multi-step reasoning and interactive decision-making, which achieves significant improvement in multiple domains. Shinn et al. (2023) develop agents that can verbally summarize from its past failures and incorporate the reflective text into subsequent trials of the same task, analogous to an online reinforcement learning paradigm. Our method, in contrast, adopts an offline learning method to minimize the cost of online learning and can adapt to various new tasks. Two concurrent works, ExpeL Zhao et al. (2024) and AutoGuide Yang et al. (2023), also perform learning from offline experiences. ExpeL only extracts tips from offline data and provides them all in the prompt, while AutoGuide extracts tips conditioning on a summarized state and performs state-based similarity retrieval to select tips. Our approach differs from theirs along three dimensions: a) we discover skills from offline experiences and extract skill-oriented tips; b) we discover primitive actions for each skill to reduce the chance of action hallucination; c) our method can also improve code-generation quality while theirs only consider text-based outputs. *(2) Combined with planning.* LLMs can also be combined with classical planning approaches, such as Planning Domain Definition Language (PDDL) (Liu et al., 2023a; Silver et al., 2023). Differently, our paper focuses on end-to-end LLM policies without additional planning algorithms or knowledge. *(3) Generating code.* Recent works showing that embodied agents can be empowered by LLM-written code (Liang et al., 2022; Wang et al., 2023a;b). Although the main focus of this paper is LLM agents interacting with environments by language, the proposed method naturally extends to code-based policies.

**Training or fine-tuning LLMs for sequential decision-making.** Another line of related work includes training textual policies with imitation learning (Shridhar et al., 2021; 2020) and fine-tuning language models to behave as policies (Wang et al., 2023a; Driess et al., 2023; Wang et al., 2023c) in sequential decision-making. Again, our work is different as it aims at achieving high-quality LLM-based decision-making without fine-tuning.

**Multi-step reasoning and task decomposition with LLMs.** Multi-step reasoning using LLMs has been widely studied in language domains, including chain-of-thought style reasoning Yao et al. (2023a); Fu et al. (2022); Wei et al. (2022), step-by-step feedback based reasoning Lightman et al. (2023); Zheng et al. (2023), and self consistency and verification based reasoning Ling et al. (2023); Wang et al. (2022). In contrast, we focus on sequential decision-making tasks in partially observable environments, where each step results in state transitions and only sparse rewards are available.

# 3 Methodology

## 3.1 Problem Formulation and Notations

**LLM-powered policy.** We first formally define an LLM-powered policy. A policy $\pi$ for sequential decision-making is a function that maps the interaction history to a distribution over actions. Let $\pi(a|\tau)$ denote the probability of selection action $a$ given interaction history $\tau$, which is a sequence of all past observations and actions, $\langle o_1, a_1, o_2, a_2, \cdots, o_t \rangle$. Then, with a pre-trained LLM which outputs text based on any text input, a policy can be written as

$$\pi_{\mathrm{llm}}(a|\tau) := LLM(a|\tau; \theta_{\mathrm{pmt}}, \theta_{\mathrm{pret}}). \tag{1}$$

The goal is to learn a policy that can maximize the total reward. During in-context learning, the pre-trained LLM weights $\theta_{\mathrm{pret}}$ are fixed, and our O3D learns a policy by learning and optimizing the base prompt $\theta_{\mathrm{pmt}}$ from an offline dataset containing multiple interaction trajectories/histories.

**Skill-conditioned policy.** A policy can be conditioned on specific skills or subgoals (e.g., find a mug)(Xu et al., 2023), which are compositional factors of the original task (e.g., heat some milk). Let $z$ be a skill, then a skill-conditional policy can be denoted as $\pi^z$, with $\pi^z(a|\tau)$ the probability of selecting action $a$ given history $\tau$ when executing skill $z$.

## 3.2 Learning from Offline Data

In many real-world decision-making systems, there exists interaction logs from various users, including experts who can successfully perform the task, as well as non-experts who may fail and make mistakes. Although a final success/failure flag is given for each trajectory, there is no intermediate rewards within the trajectory, making policy learning more difficult.

Intuitively, seeing the interaction logs from others performing a task can be helpful for one to understand the environment and finish similar tasks. Since LLMs have strong abilities of interpretation and generalization, recent works such as ReAct (Yao et al., 2023b) have shown that LLMs can solve many interactive decision-making problems when prompted with a few expert demonstrations. This learning paradigm is analogous to behavior cloning (Torabi et al., 2018), where an agent learns to imitate how experts react to certain scenarios. However, behavior cloning could suffer from the distribution shift between expert demonstrations and the agent's own online interactions, especially when the expert dataset is small and not representative of all scenarios in the domain. Even if we increase the number of demonstrations, the performance is constrained by the limited context window and the high cost of collecting expert demonstrations and annotations (e.g., write reasoning examples). In contrast, we introduce an offline learning framework for LLM-powered policies which transfers knowledge across tasks and makes use of large amounts of sub-optimal interaction log to improve downstream decision-making.

## 3.3 O3D: LLM-Based Offline Policy Improvement

Our proposed offline policy learning framework consists of 3 stages, as depicted in Figure 1. The first stage enables the LLM to discover and abstract reusable skills from offline datasets (potentially from diverse tasks). Then, the second stage aims to learn a skill-based policy for each discovered skill, through an iterative process of distilling primitives and tips to achieve policy improvement. The final stage is to construct the main LLM-powered agent who can reason and call corresponding skills sequentially to solve given tasks. Below we explain each stage in detail.

**Stage 1: Offline Skill Discovery and Data Segmentation.** Many real-world decision-making processes require a number of steps to complete a task, such as controlling a robot to pass several obstacles and navigate to the door, which results in two challenges for LLM-powered agents. First, the limited context length may not be enough to contain the few-shot demonstration and online interaction history. Second, the language model may lose track of its goal and not pay attention to the most important information. To mitigate this issue, we propose a hierarchical policy learning framework that can iteratively extracts skills from

interaction logs with primitive-level executions. Here the skills are analogous to the options or temporally extended actions (Sutton et al., 1999) in hierarchical reinforcement learning. It is well-known that discovering options is difficult in traditional RL, whereas we find that skill discovery with textual logs can be surprisingly well-achieved with the semantic understanding ability of state-of-the-art LLMs. Our skill discovery process iterates over the offline trajectories, using a DiscoverPrompt as shown in Fig. 1 (Stage 1). The full prompt we use is in Appendix E. We ask LLMs to divide the interaction histories into skill-oriented sub-trajectories, and abstract the skills in function forms.

**Stage 2: Offline Policy Improvement with Knowledge Distillation.** The main idea of this stage is to distill generalizable knowledge from offline datasets. Such knowledge should be generalizable to tolerate the distribution shift between offline data and online interactions and facilitate downstream tasks. We aim to distill two types of knowledge from the segmented skill-based trajectories in an iterative process, which leads to improved skill-conditioned policies.

• *Distilling Primitive Actions.* A common mistake of LLM-powered agents is hallucination, i.e., LLMs output actions that are not valid in the environment. To ensure effective usage of LLM in decision-making applications, it is important to specify an action space in the form of natural language or code. Prior works (Liang et al., 2022; Liu et al., 2023c) manually define the available primitive functions, which requires human labor and domain knowledge. Instead, we propose to distill primitive actions or functions from the offline interaction data with LLM, which is easy to scale up and automate the practical operation. Fig. 1 describes how to distill the primitives with an example, and the full prompt is in Appendix E.

• *Distilling Policy Improvement Tips with Trajectory Contrasting.* Inspired by the policy gradient methods in RL, which increases the probability of selecting good actions and lowers the probability of selecting bad ones, we propose to distill knowledge that can enhance good (i.e., achieving high long-term reward) behaviors and avoid undesired ones in the task distribution. We propose to distill "policy improvement tips" about what actions are preferred under what circumstances. However, with the offline data that only provides sequences of interactions and final scores, it is non-trivial for an LLM to figure out the correct credit assignment and useful tips to guide policy improvement. To this end, we leverage *Trajectory Contrasting* to sample both successful (high-score) and failed (low-score) trajectories and let the LLM generate tips by comparing them. LLMs can identify the key to success and how to avoid failures. Fig. 1 shows the iterative distillation process with a DistillPrompt to update the policy. More details are in Section 3.4 and Appendix E.

**Stage 3: Downstream Interaction with Hierarchical Policy Execution.** So far, Stage 1 discovers a set of skills, while Stage 2 produces and optimizes the corresponding skill-conditioned policies. The final stage is then to compose these skills and interact with the downstream task. We prompt a base policy $\pi_{base}$ with a few examples (come from LLM's own segmentation of offline trajectories) on calling skills sequentially given a downstream task. Therefore, the policy is executed in a hierarchical manner as Figure 1 shows: the high-level base policy receives the

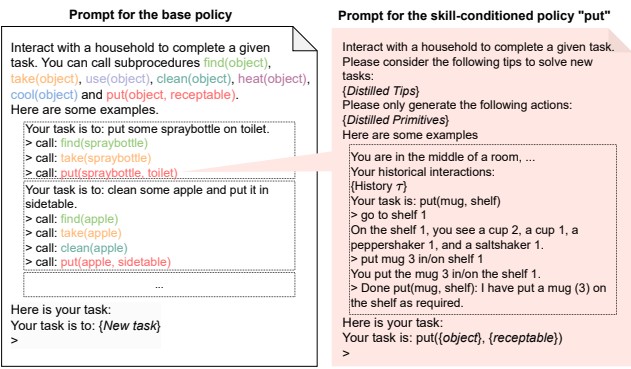

Figure 2: Example prompts for the base policy and the text-based skill-conditioned policy for hierarchical policy execution in Stage 3.

environment and calls the low-level skill-conditioned policies; the called skill-conditioned policy then interacts with the environment and returns a terminate signal when its job is done; the high-level base policy receives the update from the low-level policy and determines whether to execute the next skill. Fig. 2 shows an example of how to construct the base policy by prompting and how a skill-conditioned policy is prompted when being called. Appendix D.3 provides a concrete example of the execution process.

### 3.4 Implementation Details of O3D

The main algorithm is presented in Algorithm 1. We provide all used prompts and additional implementation details in Appendix E. In Stage 1 (Line 1-5), we traverse the offline trajectories and ask LLM to summarize potential sub-procedures as skills. Then, based on the discovered skill set, we ask an LLM to segment each trajectory into sub-trajectories, each of which corresponds to a specific skill (Line 5).

In Stage 2 (Line 6-13), the goal is to establish a set of skill-conditioned policies. For each skill, we sample a mini-batch of its (sub-)trajectories, then discover primitive actions and distill policy improvement tips. Stage 3 (Line 14-15) is the downstream inference process, where the agent observes the task and sequentially calls the learned skill-conditioned policies to interact with the environment, with few-shot examples of how to call the skills.

The details of an extension of the above method, O3D-Code, for code-generation and empirical results are in Appendix D, where O3D-Code outperforms existing methods by learning from offline trajectories.

---

**Algorithm 1** Offline Data-driven Discovery and Distillation

**Input:** Pre-trained LLM, offline dataset $\mathcal{D}$, batch sizes $N_1, N_2$, max iteration steps $T_1, T_2$
**Output:** LLM-powered policy $\pi$
`// Stage 1: skill discovery`
1  Initialize sets of skills $\mathcal{Z} = \varnothing$
2  **for** $t = 1, \ldots, T_1$ **do**
3      Sample $N_1$ trajectories $d \sim \mathcal{D}$
4      Discover skills:
         $\mathcal{Z} \leftarrow LLM(\mathcal{Z}, d; \text{DiscoverPrompt})$
5  Segment trajectories in $\mathcal{D}$ based on skillset $\mathcal{Z}$, obtain $\mathcal{D}^{z_k}$ for each $z_k \in \mathcal{Z}$
`// Stage 2: knowledge distillation`
6  **for** $z_k \in \mathcal{Z}$ **do**
7      Initialize primitive set $\mathcal{P}^{z_k}$ and the knowledge set $\mathcal{T}^{z_k}$
8      Initialize $\pi^{z_k} \leftarrow$ Initiate $\theta_{\text{pmt}}$ with $d \sim \mathcal{D}^z$ and $\mathcal{P}^z$
9      **for** $t = 1, \ldots, T_2$ **do**
10         Sample $N_2$ trajectories $d^{z_k} \sim \mathcal{D}^{z_k}$
11         Discover primitives:
            $\mathcal{P}^{z_k} \leftarrow LLM(\mathcal{P}^{z_k}, d^{z_k}; \text{DiscoverPrompt})$
12         Distill policy improvement tips:
            $\mathcal{T}^{z_k} \leftarrow LLM(\mathcal{T}^{z_k}, d^{z_k}; \text{DistillPrompt})$
13         Ask LLM to incorporate $\mathcal{T}^z$ into the prompt of policy $\pi^z$
`// Stage 3: hierarchical policy execution`
14 Sample examples $d \sim \mathcal{D}$ and segment them based on skills
15 Provide the examples as demonstrations for $\pi_{\text{llm}}$ to call skill-conditioned policies $\{\pi^z\}_{z \in \mathcal{Z}}$ given downstream tasks

---

## 4 Experiments

### 4.1 Experimental Setup

**Problem Domains.** We consider two sequential decision-making benchmarks, ALFWorld (Shridhar et al., 2021) and WebShop (Yao et al., 2022). ALFWorld mimics household scenarios and allows an agent to interact with the environment through a text-based interface We use the original test set, including 134 tasks in total across six distinct task types. Following Shinn et al. (2023), we make the problem more challenging by limiting the horizon of each episode to be 30 (original is 50) steps and terminating the episode if the agent takes the same action twice. WebShop requires an agent to explore an online shopping website, check relevant product candidates, and purchase the one that matches a user's query. Our evaluation considers the first 500 out of 12,087 instructions as test set (following the official implementation (Yao et al., 2022)).

**Metrics.** In ALFWorld, we assess method performance by measuring the success rate (SR) under each task type and a total success rate over 134 tasks. Besides the success rate, WebShop sets a product matching score as an extra metric.

**Models and Offline Data.** To investigate the robustness of O3D across a range of LLMs, we consider two different GPT models in our experiments. These include GPT-4-0613 and GPT-3.5-0613, each providing distinct $\theta_{\text{pret}}$ defined in Equation (1). Our offline dataset comprises official human demonstrations in both domains as the success data, and a set of failure data generated by running ReAct on the training task set, as introduced in the original ALFWorld and WebShop implementations (more details are referred to Appendix A.1).

**Baselines and Experiment Organization.** We evaluate and analyze the proposed O3D from multiple angles by comparison with several state-of-the-art baseline methods and their variants. We first compare the downstream performance of O3D with a well-known baseline ReAct (Yao et al., 2023b). The results demonstrate the capability of O3D to discover and distill knowledge from offline dataset and improve downstream task-solving. Then, we compare O3D with an online learning algorithm Reflexion (Shinn et al., 2023) which interacts with each

downstream task for multiple trials and improves the policy by verbally reflecting on its failure history. We also evince the versatility of `O3D` by showing the advantage of combining `O3D` with existing methods (Reflexion+O3D). Furthermore, we conduct comprehensive ablation study for each component of `O3D` including primitives, skills, policy improvement tips, and trajectory contrasting. We also show the comparison with ReAct-long, which inserts as many offline trajectories to the prompt context as possible, to further validate the efficiency of our distillation mechanism.

## 4.2 Main Results

**Effectiveness of `O3D` in discovering skills and primitives, and distilling policy improvement tips.** In our experiments, `O3D` can extract high-quality skills from raw successful trajectories in offline data, resulting in seven types of skills under ALFWorld domain, including: *find(object)*, *take(object)*, *put(object, receptacle)*, *cool(object)*, *heat(object)*, *use(object)* and *clean(object)*; and four types of skills under WebShop domain, including *search item*, *select item*, *select item's attributes* and *purchase item*. Each skill consists of a set of primitives for execution, accompanied by a group of tips to facilitate skill completion. Fig 3 shows four skill examples from the ALFWorld domain (full results are available in Appendix E.4). Through learning from offline data, `O3D` is able to capture correct primitives that can be composed to achieve each corresponding skill. Importantly, we observe that tips distilled by `O3D`

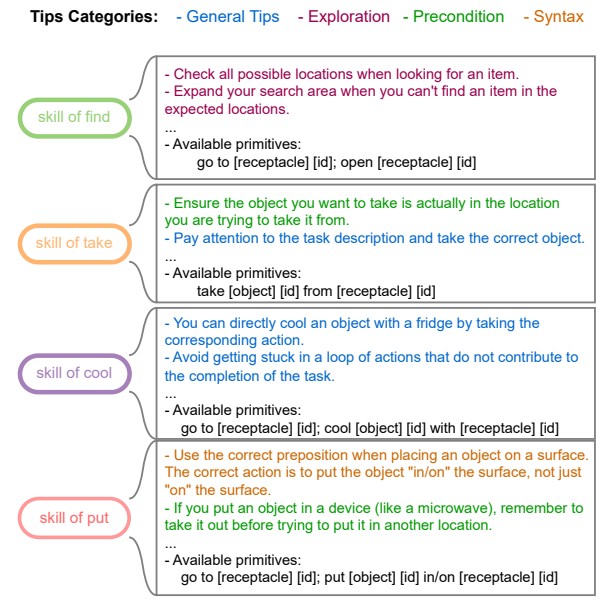

Figure 3: Examples of discovered skills and distilled knowledge and tips in ALFWorld.

are functional to various degrees, such as suggesting general tips, encouraging exploration, realizing action preconditions and highlighting syntax (as shown in Fig. 3). To further assess the quality of LLM-distilled knowledge, we designed an oracle baseline, `O3D-Human`, where a domain expert writes down tips and primitives in the prompt according to ReAct's failures. The comparative results in Appendix **??** demonstrate that `O3D` is promising to extract human-level knowledge from offline data such as achieving the performance near `O3D-Human`, and even surpassing `O3D-Human` in certain cases.

Table 1: Main results on ALFWorld and WebShop in comparison to ReAct.

(a) Results in ALFWorld

| Model | Method | Pick | Clean | Heat | Cool | Look | Pick2 | All |
|-------|--------|------|-------|------|------|------|-------|-----|
| GPT-4 (0613) | ReAct | 67 | 74 | 74 | 67 | 100 | 47 | 72 |
|  | O3D | **92** | **100** | **96** | **95** | **100** | **53** | **91** |
| GPT-3.5 (0613) | ReAct | 13 | 10 | 0 | 0 | 17 | 0 | 7 |
|  | O3D | **71** | **35** | **4** | **67** | **44** | **24** | **41** |

(b) Results in WebShop

| Model | Method | SR | Score |
|-------|--------|----|----|
| GPT-4 (0613) | ReAct | 26 | 39 |
|  | O3D | **41** | **58** |
| GPT-3.5 (0613) | ReAct | 27 | 60 |
|  | O3D | **35** | **61** |

`O3D` **consistently outperforms baselines across various LLMs under ALFWorld** (Table. 1a) and **WebShop** (Table. 1b). `O3D` achieves higher success rates than ReAct by 19% and 34% in ALFWorld, and 15% and 8% in WebShop, with using GPT-4-0613 and GPT-3.5-0613 respectively. Furthermore, as shown in Table 1a, the success rate achieved by `O3D` in each task category consistently exceeds that of ReAct with the two GPT models. This further confirms that the tips distilled by O3D from offline data are generalizable and beneficial

Table 2: Comparison with online-learning method Reflexion and a combination between Reflexion and O3D.

(a) Results in ALFWorld

| Model | Method | Pick | Clean | Heat | Cool | Look | Pick2 | All |
|---|---|---|---|---|---|---|---|---|
| GPT-3.5 (0613) | Reflexion | 33 | 26 | 26 | 24 | 50 | 18 | 29 |
| | O3D | **71** | 35 | 4 | 67 | 44 | 24 | 41 |
| | Reflexion+O3D | 54 | **35** | **61** | 67 | **56** | **29** | **50** |

(b) Results in WebShop

| Model | Method | SR | Score |
|---|---|---|---|
| GPT-3.5 (0613) | Reflexion | 35 | 55 |
| | O3D | 35 | 61 |
| | Reflexion+O3D | **46** | **62** |

across diverse task types. For example, the tip "pay attention to the task description and take the correct object" helps the LLM agent avoid taking a similar object (a pot) rather than the requested one (a pan); and the tip "The correct action is to put the object 'in/on' the surface, not just 'on' the surface" prevents the LLM agent from making syntactical errors with primitives, which are the two common mistakes made by ReAct.

We also observe that, the overall performance of GPT-3.5-0613 is significantly lower than GPT-4-0613 in the ALFWorld domain. Similar poor performance on ALFWorld is also reported by the AgentBench paper Liu et al. (2023b). This discrepancy may be attributed to the fact that GPT-3.5-0613 is fine-tuned with data and objectives that are diverged from the ALFWorld domain. Appendix D.3 provide some interaction logs on these tasks and analyze the potential causes of the failure. Despite the misalignment of GPT-3.5-0613 with the ALFWorld domain, our approach still achieves 34% improvement over ReAct.

**Advantages of offline learning over online iterative reflection.** We compare the performance of O3D with Reflexion (Shinn et al., 2023) that involves 10 and 3 iterations of online reflection in ALFWorld and WebShop respectively (see Table 2a and Table 2b). Reflexion allows an LLM agent to reflect on its online interaction history via multiple iterations for performance improvement. However, such online interactions in a real-world environment can be expensive or risky. In contrast, O3D learns from offline data, which may include interaction logs from various users or agents. Remarkably, O3D outperforms Reflexion in its first deployment trial without additional online reflection. More importantly, Reflexion learns task-specific textual tips, which only works for the current task it is solving. In contrast, O3D identifies reusable skills and generalizable textual tips that can facilitate a wide range of downstream tasks. Also, Reflexion relies on online failure experiences, however, O3D offers leverages both successful and failed experiences for improvements, which is more effective in generating proper tips to address mistakes.

O3D **can be combined with existing methods to enhance their performance.** The comparison between Reflexion and Reflexion+O3D validates that, our offline learning framework can improve existing methods by prepending the discovered primitives and distilled tips from offline data into their prompts. The results in Table 2a and Table 2b show that O3D significantly boosts Reflexion's performance under both ALFWorld and Webshop domains.

Table 3: Comparison with ReAct-long which naively injects demonstrations into prompts.

(a) Results in ALFWorld

| Model | Method | Pick | Clean | Heat | Cool | Look | Pick2 | All |
|---|---|---|---|---|---|---|---|---|
| GPT-3.5 (0613) | ReAct | 13 | 10 | 0 | 0 | 17 | 0 | 7 |
| | ReAct-long | 71 | 6 | 0 | 5 | 17 | 24 | 21 |
| | O3D | **71** | **35** | **4** | **67** | **44** | **24** | **41** |

(b) Results in WebShop

| Model | Method | SR | Score |
|---|---|---|---|
| GPT-3.5 (0613) | ReAct | 27 | 60 |
| | ReAct-long | 30 | **64** |
| | O3D | **35** | 61 |

## 4.3 Ablations

O3D **is much more effective than naively injecting demonstrations into prompts.** To demonstrate this, we add the same set of offline trajectories into the original ReAct's prompt as many as we can, with GPT-3.5-0613-16k, which is referred to as ReAct-long. As results shown in Table 3a and Table 3b, ReAct-long demonstrated improved performance over the original ReAct because of the richer context from the offline trajectories. Yet, O3D's outstanding performance in ALFWorld highlights its superior effectiveness in skill discovery and knowledge distillation from offline data. In Webshop, O3D reaches a higher success rate but a lower score than ReAct-long, because it focuses on distilling tips to purchase the item exactly matching the instruction via success-failure trajectories contrasting, rather than

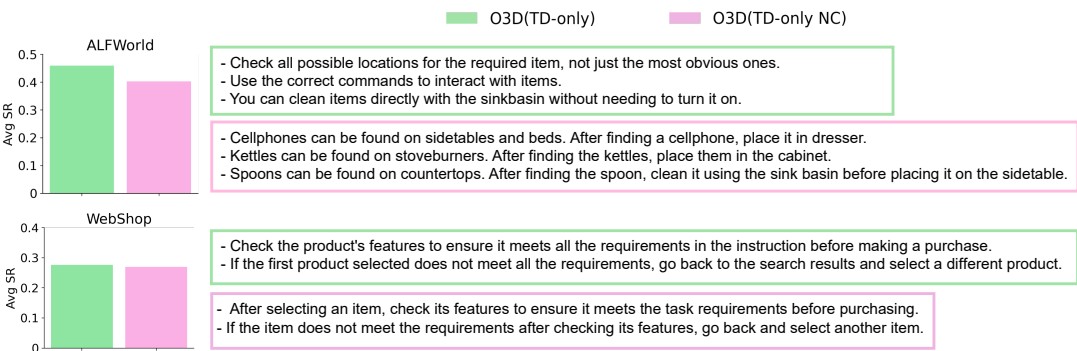

Figure 5: Comparison on averaged success rate over GPT models using contrastive and non-contrastive (NC) distillation methods, with tips in green and pink boxes respectively.

increasing the raw score. By having the full trajectories with scores, however, ReAct-long has a more fine-grained understanding of the relation between tasks and scores. A potential improvement for O3D is to contrast trajectories based on a range of scores, rather than just success or failure. Besides, ReAct-long's long prompt would incur much higher API calling costs for a large number of downstream tasks.

**Primitives, skills and policy improvement tips independently advance baseline performance.** `O3D` has three major processes: skill discovery (SD), primitives discovery (PD) and policy improvement tip distillation (TD). To investigate each component's contribution to performance improvement, we conducted an ablation study considering three variants of `O3D`, each with only one component.

Fig. 4 shows the three variants of `O3D` either outperform the baseline or achieve the same performance as the baseline over two LLM models. In ALFWorld, `O3D (PD-only)` plays the dominant role in performance improvement with GPT-4-0613, because the major mistakes made by the baseline are outputting primitive actions with syntax errors or hallucinating unavailable actions. `O3D (SD-only)` boosts the performance the most with GPT-3.5-0613, because the tasks in ALFWorld are too complex for ReAct with GPT-3.5-0613, and `O3D (SD-only)` solves the tasks in hierarchy by performing skill selections that greatly reduces the complexity. In WebShop, the three components consistently benefit the baseline performance across the two GPT models, with their individual contributions also being model-dependent. Since the offline data was collected using GPT-3.5-0613, we observe that the highest overall improvement of the three components occurs in this model.

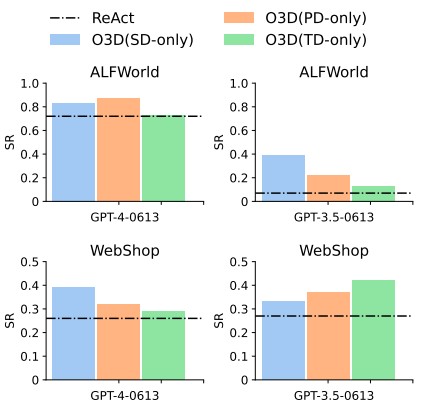

Figure 4: Comparison on success rate (SR) with three variants of `O3D` against ReAct.

**Trajectory contrasting produces domain-dependent advantages over non-contrasting approach.** Fig. 5 shows that contrasting successful and failed trials in offline data is relatively beneficial in certain domains, compared to the non-contrastive (NC) way using only successful data. In ALFWorld, failures often stem from disregarding domain-specific dynamics and rules, and the contrastive method effectively produces generic tips (green box in Fig. 5) to correct mistakes in failure cases. Conversely, the non-contrastive method merely summarizes the facts (pink box in Fig. 5) from successful trials, offering limited utility. In WebShop, however, there is often no clear mistake, such as different searching keywords may result in the same products and multiple items fitting the task requirements. Therefore, the advantage of contrastive method is slight, and both methods output analogous tips as shown in Fig. 5.

## 5 Conclusion

This paper introduces an offline in-context learning framework, O3D, for sequential decision-making, where LLM agents can learn from previous experiences in a scalable offline manner to improve performance without model fine-tuning. O3D stands out by allowing LLMs to distill shared high-level knowledge from offline interaction logs, which is injected into a single set of prompts to be reused in solving diverse downstream tasks. Empirically, O3D outperforms baseline methods in two challenging benchmark domains, and it is compatible with existing LLM-based methods for sequential decision-making. Our work offers a possibility for future LLM-based algorithm development in terms of efficiently leveraging offline data at scale for real-world sequential decision-making applications.

**Disclaimer**. This paper was prepared for informational purposes in part by the Artificial Intelligence Research group of JPMorgan Chase & Co 'and its affiliates ("JP Morgan"), and is not a product of the Research Department of JP Morgan. JP Morgan makes no representation and warranty whatsoever and disclaims all liability, for the completeness, accuracy or reliability of the information contained herein. This document is not intended as investment research or investment advice, or a recommendation, offer or solicitation for the purchase or sale of any security, financial instrument, financial product or service, or to be used in any way for evaluating the merits of participating in any transaction, and shall not constitute a solicitation under any jurisdiction or to any person, if such solicitation under such jurisdiction or to such person would be unlawful.

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

# Appendix

## A   Details of Implementation and Extra Results

### A.1   Offline Data Collection and Usage

The offline dataset used in our experiment consists of both success and failure interaction trajectories. The success data in ALFWorld are gathered from a 'traj.json' file in the training dataset provided on their official website. We download human demonstrations from the official website of WebShop, and we select out the demonstrations with 100% product matching score from the training dataset as the success data. We generate the failure data by running ReAct with GPT-3.5-0301 on ALFWorld and running ReAct with GPT-3.5-0613 on the WebShop, which are both conducted on the training task set. Eventually, we have totally 80 pairs of success trajectories and failure trajectories on the same tasks in ALFWorld and 90 pairs of success trajectories and corresponding failure trajectories in WebShop.

In Table 5, we detail the usage of the offline data for discovering primitives and skills and distilling policy improvement tips. We use GPT-4-0613 for all discovery and distillation processes. In our experiments, we note that the batch size may impact the quality of results. The tips distillation in O3D is skill-oriented, where we need contrastive data pairs for each skill. Among the first 30 offline data pairs, certain skills have zero failure data (e.g., cool) as the mistakes happened in other skills. We, therefore, generate more offline data (80) for the tips distillation part of O3D.

Table 4: Details of offline data usage in ALFworld

| Method | Component | Data Type | Batch Size | # of Trajs |
|---|---|---|---|---|
| O3D | Skill Discovery | success | 1 | 6 |
| | Primitives Discovery | success | 1 | 12 |
| | Tips Distillation | success & failure | 1 | $80 \times 2$ |
| O3D (SD-only) | Skill Discovery | success | 1 | 6 |
| O3D (PD-only) | Primitives Discovery | success | 1 | 12 |
| O3D (TD-only) | Tips Distillation | success & failure | 2 | $30 \times 2$ |
| O3D (TD-only NC) | Tips Distillation | success | 2 | 30 |

Table 5: Details of offline data usage for in WebShop

| Method | Component | Data Type | Batch Size | # of Trajs |
|---|---|---|---|---|
| O3D | Skill Discovery | success | 2 | 30 |
| | Primitives Discovery | success | 1 | 90 |
| | Tips Distillation | success & failure | 1 | $30 \times 2$ |
| O3D (SD-only) | Skill Discovery | success | 2 | 30 |
| O3D (PD-only) | Primitives Discovery | success | 1 | 90 |
| O3D (TD-only) | Tips Distillation | success & failure | 1 | $30 \times 2$ |
| O3D (TD-only NC) | Tips Distillation | success | 1 | 30 |

## B Comparison between LLM-Generated Knowledge and Human-Generated Knowledge

Table 6: Comparison with O3D-Human in ALFWorld and WebShop.

(a) Results in ALFWorld

| Model | Method | Pick | Clean | Heat | Cool | Look | Pick2 | All |
|---|---|---|---|---|---|---|---|---|
| GPT-4 | O3D | **92** | **100** | **96** | **95** | **100** | 53 | **91** |
| (0613) | O3D-Human | 83 | 100 | 87 | 95 | 100 | 53 | 88 |
| GPT-3.5 | O3D | **71** | 35 | 4 | 67 | **44** | 24 | 41 |
| (0613) | O3D-Human | 71 | **68** | **83** | **71** | 44 | 24 | **63** |

(b) Results in WebShop

| Model | Method | SR | Score |
|---|---|---|---|
| GPT-4 | O3D | **41** | 58 |
| (0613) | O3D-Human | 41 | **61** |
| GPT-3.5 | O3D | **35** | 61 |
| (0613) | O3D-Human | 31 | 61 |

Table 7: Tips of the "*put(object. receptacle)*" skill in ALFWorld

| Method | Tips |
|---|---|
| LLM | 1. Use the correct preposition when placing an object on a surface. The correct action is to put the object "in/on" the surface, not just "on" the surface.
2. Place the object in the first available and suitable location instead of unnecessarily checking other locations.
3. If you put an object in a device (like a microwave), remember to take it out before trying to put it in another location. |
| Human | 1. You can directly put an object in/on an occupied receptacle.
2. Please strictly follow the syntax of primitives. |

Table 8: Knowledge of the "*select item's attributes*" skill in WebShop

| Method | Tips | Primitives |
|---|---|---|
| LLM | 1. Ensure to select all the necessary attributes as per the instruction.
2. Be aware that color and size options may not always be straightforward and may include additional information or codes.
3. Be careful with the case and spacing of the words when selecting item attributes. | click[Attribute]
click[Color]
click[Style]
click[Size]
click[Flavor] |
| Human | 1. Remember to select all required attributes using click[Attributes] before click[Buy Now].
2. You should pay attention to the current observation to check clickable buttons. | click[Attribute] |

**O3D is promising to extract human-level knowledge from offline data such as achieving the performance near O3D-Human, and even surpassing O3D-Human in certain cases.** O3D-Human is designed as an oracle, where a domain expert writes down tips and primitives in the prompt according to ReAct's failures. We noticed that human often gives more general tips/primitives, while LLMs can distill both general and granulated ones because of the contrastive way over success-failure pairs. Given the same sets of LLM-generated knowledge and human-generated knowledge to three different models, as Table 6 shown, which type of knowledge is better depends on the model's capability, such as O3D outperforming O3D-Human when using GPT-4 in ALFWorld and GPT-3.5-0613 in WebShop, but being relatively worse than O3D-Human on other models. Please see the concrete examples in Table. 7 and Table. 8, where the LLM-generated knowledge is more comprehensive than those generated by humans.

In Table. 7, the LLM-generated tips are more comprehensive such as the concrete in-line examples of obeying action syntax; the suggestion for avoiding redundant behaviors; and the

hint on the action precondition. We think this is why `O3D` always outperformed `O3D-Human` on the "Pick" task across three GPT models.

In Table. 8, the LLM-genreated tips offer detailed guidance on aspects to pay attention during attribute selection, along with more granular primitives being discovered to assist the agent in focusing on various types of attributes.

Overall, the comparison between `O3D` and `O3D-Human` verifies that LLMs can distill tips and discover primitives as good as human-generated ones. Please see Appendix E.3 and Appendix E.4 for the full tips and primitives of human-generated and LLM-generated.

## C  Additional Results

In this section, we present additional experimental results in ALFWorld and WebShop domains.

### C.1  ALFWorld

In our ablation study, we also further visualize the performance of the three variants of `O3D` under each type of task with three different models, as shown in Fig 6. From the perspective of each task type, the advantage of the three variants of `O3D` versus ReAct is a bit model-dependent, but overall, they outperform ReAct as shown in the last column of Fig. 6.

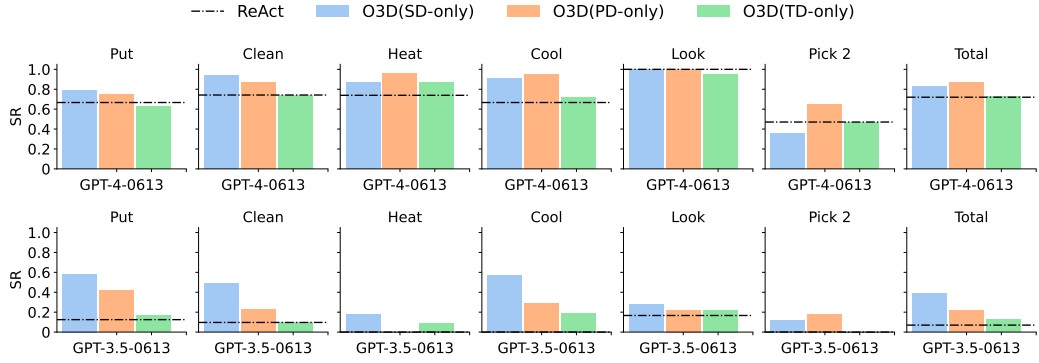

Figure 6: Comparison on success rate (SR) with three variants of `O3D` against ReAct.

### C.2  WebShop

We show the performance of the three variants of `O3D` on the product matching score against the baseline in Fig 7. With GP4-0613 and GPT-3.5-0613, the three variants of `O3D` all outperform ReAct, which validates the contribution of the offline data-driven discovery and distillation to policy improvement.

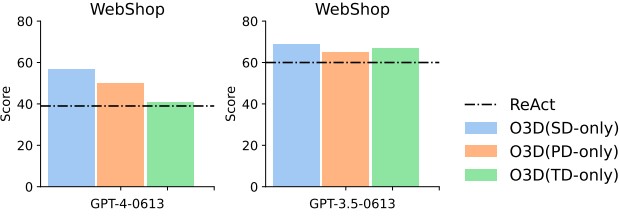

Figure 7: Comparison on the product matching score with three variants of `O3D` against ReAct.

# D Extension of `O3D` to Code-Generation

## D.1 Methodology

This paper mainly focuses on Text-based-Policy which takes in textual observations from the environment and outputs textual actions to interact with the environment. But the proposed approach is also applicable to other scenarios of LLM-environment interaction. For example, it is well-known that LLMs can write code to solve tasks, where one asks LLMs to generate code and implement the policy function given proper prompts, i.e.,

• *Code-based-Policy.*
$$\pi_{\text{code}}(a|\tau) := Code(a|\tau) \leftarrow LLM(\theta_{\text{pmt}}, \theta_{\text{pret}}). \tag{2}$$

In this case, the goal of learning is to optimize the produced function code $Code(a|\tau)$ without tuning the pre-trained weights $\theta$. The central idea of `O3D` naturally applies to Code-based-Policy: the LLM discovers reusable skills (callable functions) and distills policy improvement tips from offline dataset; the obtained knowledge helps LLM refine its generated code policy iteratively; during execution stage, the base policy sequentially determines which skills/functions to call, and then the corresponding skill function is executed.

---

**Algorithm 2** Learning Code-based-Policy with Offline Data-driven Discovery and Distillation (`O3D`)

---

**Input:** Pre-trained LLM, offline dataset $\mathcal{D}$, batch sizes $N_1, N_2$, max iteration steps $T_1, T_2$
**Output:** Code-based-Policy $\pi$
`// Stage 1: skill discovery`
16 Initialize sets of skills $\mathcal{Z} = \varnothing$
17 **for** $t = 1, \ldots, T_1$ **do**
18     Sample $N_1$ trajectories $d \sim \mathcal{D}$
19     Attempt to discover more skills $\mathcal{Z} \leftarrow LLM(\mathcal{Z}, d; \text{DiscoverPrompt})$
20 Segment trajectories in $\mathcal{D}$ based on skillset $\mathcal{Z}$, obtain $\mathcal{D}^{z_k}$ for each $z_k \in \mathcal{Z}$
`// Stage 2: knowledge distillation`
21 **for** $z_k \in \mathcal{Z}$ **do**
22     Initialize primitive set $\mathcal{P}^{z_k}$ and the knowledge set $\mathcal{T}^{z_k}$
23     Initialize $\pi^{z_k} \leftarrow$ Let LLM write a function to reproduce $d \sim \mathcal{D}^z$ with primitive functions $\mathcal{P}^z$
24     **for** $t = 1, \ldots, T_2$ **do**
25        Sample $N_2$ trajectories $d^{z_k} \sim \mathcal{D}^{z_k}$
26        Attempt to discover more primitives $\mathcal{P}^{z_k} \leftarrow LLM(\mathcal{P}^{z_k}, d^{z_k}; \text{DiscoverPrompt})$
27        Distill policy improvement tips $\mathcal{T}^{z_k} \leftarrow LLM(\mathcal{T}^{z_k}, d^{z_k}; \text{DistillPrompt})$
28        Ask LLM to improve $\pi^z$ based on suggestions $\mathcal{T}^z$
`// Stage 3: hierarchical policy execution`
29 Sample examples $d \sim \mathcal{D}$ and segment them based on skills
30 Provide the examples as demonstrations for $\pi_{\text{llm}}$ to call skill-conditioned policies $\{\pi^z\}_{z \in \mathcal{Z}}$ given downstream tasks

---

Algorithm 2 illustrates the implementation of Code-based-Policy. The major algorithm flow is similar to the text-based one in Algorithm 1, with several distinctions below:
• *Policy Initilization with Primitives.* Text-based-Policy directly provides the discovered primitives in the prompt of policy and advises the agent to follow the primitives, while Code-based-Policy first lets the LLM write primitive functions and then calls these primitive functions in the code of skill-conditioned policies.
• *Policy Improvement.* Since the distilled policy improvement tips are in natural language, we directly ask the LLM to merge the new suggestion into the prompt of Text-based-Policy. For Code-based-Policy, we let the LLM consider the policy improvement tips and re-write the policy code.
• *Policy Construction and Execution.* In Stage 3, we prompt the base policy to call the learned Text-based-Policy or Code-based-Policy. Note that for Code-based-Policy, it is possible

that the generated skill-conditioned code has compilation errors, so that it requires human checking or validation on a small set of tasks to verify that the code is executable.

Using LLMs to directly interact with environments (Text-based-Policy) and using LLMs to write code to interact with environments (Code-based-Policy) are usually discussed separately. Our study also reveals the different **pros and cons** of these two approaches that is not well-discussed in literature.
• *Advantages of Code-based-Policy.* Code-based-Policy explicitly writes the acting policy in code, which is more interpretable and reliable, and can fully avoid hallucination or syntax errors in execution. Moreover, Code-based-Policy is usually more cost efficient, as the generated code can be reused in new tasks without calling LLMs. Therefore, Code-based-Policy can be more suitable for applications where reliability and efficiency are important.
• *Advantages of Text-based-Policy.* Text-based-Policy is relatively easy to implement in practice with less human supervision. Also, in complicated environments such as WebShop where language understanding is important, Text-based-Policy can achieve much better performance than Code-based-Policy, as it retains the commonsense, expressiveness and reasoning ability of pre-trained LLMs. Therefore, for language-oriented applications where reasoning and the ability of recovering from failure are crucial, Text-based-Policy, or a combination of the two approaches, can be a better choice.

• *Advantages of Code-based-Policy.* • *Reliability and robustness.* Code-based-Policy understands the interaction logic and writes the acting policy in code, which is more interpretable and reliable, and can fully avoid hallucination or syntax errors in execution.
• *Cost and efficiency.* Code-based-Policy is usually more cost efficient, as the generated code can be reused in new tasks without calling LLMs.
• *Flexibility and expressiveness.* In more complicated environments, Text-based-Policy may achieve much better performance than Code-based-Policy, as it remains the commonsense and reasoning ability of pretrained LLMs, especially when language understanding is needed.
• *Implementation difficulty and anticipated human supervision.* Text-based-Policy is relatively easy to implement in practice with less human supervision. For Code-based-Policy, as discussed above, supervision or certain level of online validation may be required to ensure the code is executable.

## D.2    Experiments

In this section, we present additional experimental results of `O3D-Code` in ALFWorld and WebShop domains.

### D.2.1    ALFWorld

We show the comparison of `O3D-Code` against two state-of-the-art code-as-policy baselines, Demo2Code Wang et al. (2023b) and AdaPlanner Sun et al. (2023) in Table. 9. Note, we ran the official AdaPlanner code with GPT-3.5-0613. It ended up with a much worse performance than the results in the original paper. We think this is because their results were generated by GPT-3.5-0301.

The extension of `O3D` to code-generation, `O3D-Code`, demonstrates advancements comparing with two established baselines, Demo2Code Wang et al. (2023b) and AdaPlanner Sun et al. (2023), in both domains. `O3D-Code` achieves an impressive leading performance against Demo2Code by 42% with GPT-4-0613, 32% with GPT-3.5-0613, 44% with GPT-3.5-0301, and outperforms AdaPlanner by 65% with GPT-3.5-0613, 9% with GPT-3.5-0301. The principal advantage of our approach lies in its unique method of generating code: it adopts a bottom-up style, effectively constructing policies on top of robust skill functions. Through iterative skill refinement, the model cultivates robust skills by leveraging extensive and diverse data from demonstrations. Skill functions can then be efficiently utilized and reused to compose higher-level policies. In contrast, Demo2Code follows a top-down approach, requiring the generation of code for the same set of skills each time it receives a new task. Due to the context length constraint inherent in LLMs, only a limited number of demonstrations are used to guide skill code generation in Demo2Code, resulting in unstable

Table 9: Results of Code-based-Policy in ALFWorld

| Model | Method | Pick | Clean | Heat | Cool | Look | Pick2 | SR |
|-------|--------|------|-------|------|------|------|-------|-----|
| GPT-4-0613 | Demo2Code | 96 | 58 | 13 | 43 | 0 | 65 | 48 |
|  | O3D-Code | **100** | **84** | **87** | **90** | 89 | **88** | **90** |
| GPT-3.5-0613 | Demo2Code | 96 | 26 | 48 | 29 | 0 | **82** | 46 |
|  | AdaPlanner | 4 | 13 | 0 | 0 | 0 | 0 | 13 |
|  | O3D-Code | **100** | **71** | **91** | **86** | 89 | 18 | **78** |
| GPT-3.5-0301 | Demo2Code | 96 | 48 | 35 | 19 | 17 | 53 | 46 |
|  | AdaPlanner | 78 | **94** | 70 | **94** | 63 | 78 | 81 |
|  | O3D-Code | **96** | 84 | **91** | 86 | 89 | 88 | 90 |

and inconsistent skills. Although Adaplanner incorporates an online reflection process for code correction, typically over 6 iterations as its official setup, it still falls short in terms of success rate compared with O3D-Code. Notably, its performance is significantly worse than O3D-Code when utilizing GPT-3.5-0613. This comparison further confirms the superiority of O3D-Code's bottom-up way for code-generation and the iterative skill refinement over offline data.

We also conduct experiments to investigate whether the policy improvement tips distilled for text-based policy can improve the performance of code-based policy or not. As the results shown in Fig. 8, in ALFWorld, the tips distilled with the contrastive approach and the non-contrastive approach both have a minor contribution to the improvement of code-based policy.

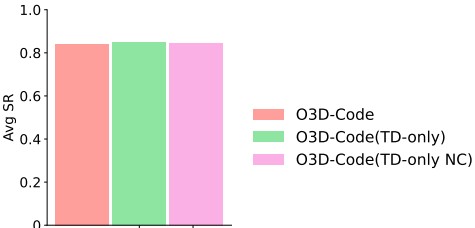

Figure 8: Results of O3D-Code with the tips distilled via contrastive and non-contrastive ways against the original O3D-Code in ALFWorld. We use the averaged success rate over the three GPT models as the performance of each method.

### D.2.2 WebShop

We show the comparison of O3D-Code against Demo2Code Wang et al. (2023b) in Table. 10. (Note that AdaPlanner does not provide a solution for WebShop, thus we did not compare with it.)

Table 10: Results of Code-based-Policy in WebShop

| Model | Method | SR | Score |
|-------|--------|-----|-------|
| GPT-4-0613 | Demo2Code | 1 | 5 |
|  | O3D-Code | **19** | **31** |

As the results shown in Table. 10, Webshop poses a substantial challenge for code-based policies owing to its need for comprehensive natural language understanding within the environment feedback. We address this challenge by enabling LLMs to construct skills by employing a limited set of LLM-based functions that encapsulate the underlying LLM capabilities (see Appendix E.6.2). To ensure a fair comparison, we also offer the same set of

functions to Demo2Code. While our approach substantially enhances the performance of code-based policies in comparison to the baseline, it's important to note that skill generation remains considerably constrained by the intricate and diverse text-based environment feedback in Webshop.

Additionally, in WebShop, as the results shown in Fig 9, the tips distilled for text-based policy is not helpful for the improvement of code-based policy.

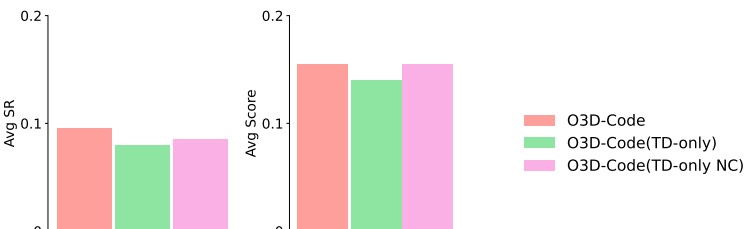

Figure 9: Results of O3D-Code with the tips distilled via contrastive and non-contrastive ways against the original O3D-Code in WebShop. We measure the averaged success rate and the averaged product matching score over GPT-4-0613 and GPT-3.5-16K-0613 as the performance of each method.

### D.3 Detailed Interaction Logs: Examples and Comparisons

In Figure 10, we visualize the downstream interaction logs of both ReAct and our O3D on the same task from ALFWorld, to better illustrate why O3D has advanced performance. We can see that ReAct agent makes a syntax mistake when executing the action of putting a potato. As a result, the invalid action does not work in the environment, but the agent does not realize the error and believes it has successfully finished the task. In contrast, O3D executes the policy in a hierarchical way, and each skill-conditioned policy works as expected. Also, the "put" agent can successfully avoid the syntax error as suggested by the distilled knowledge.

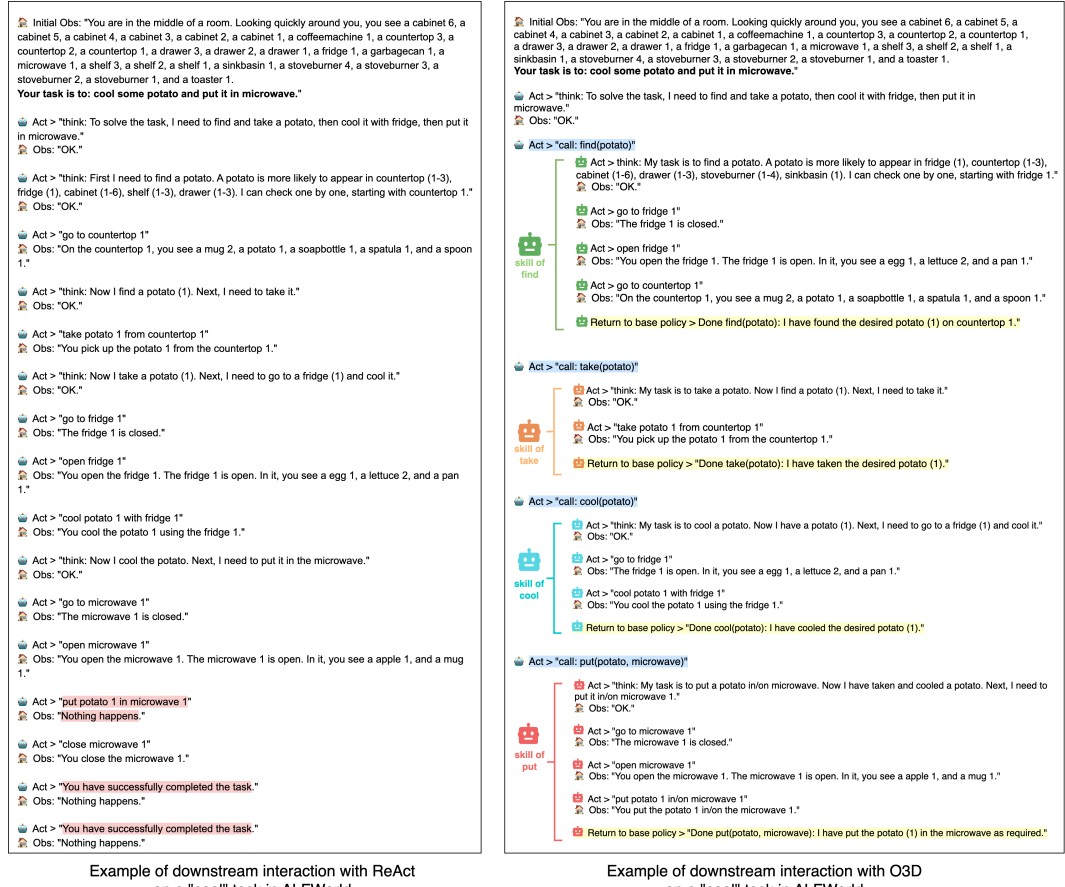

Figure 10: Interaction logs of ReAct (left) and O3D (right) in the same task with GPT-4.

In experiments, we observe that the decision-making performance can be significantly different in different GPT versions. For example, GPT-3.5-0613 struggles to solve some ALFWorld tasks (ReAct with GPT-3.5-0613 fails on all Heat and Cool tasks as shown in Table 6). To better understand this phenomenon, we examine the downstream interaction log of the agent in these tasks. Figure 11 (left) shows the expert demonstration in a task where the agent heats an apple and puts it in a fridge, which is the example provided by ReAct Yao et al. (2023b) in its base prompt. Given this example, however, we find that the ReAct agent with GPT-3.5-0613 model does not successfully solve the downstream task that is very similar to the expert example. When the agent finds that the microwave is closed, it does not follow the expert demonstration to heat the apple. Instead, it tries to open the microwave and retry going to the microwave. The agent finally gets stuck in a dead loop of going to the microwave and opening it. This is a common problem of GPT-3.5-0613 across many test tasks, showing the difficulty of letting LLM agent solve sequential decision-making problems.

Expert demonstration on a "heat" task in ALFWorld

Interaction log of ReAct on a "heat" task in ALFWorld

Figure 11: Expert demonstrations for a "heat" task in ALFWorld (left) and a failure trajectory of ReAct with GPT-3.5-0613 on a "heat" task given the expert demonstration in its prompt (right).

# E  Prompts and Learned Knowledge

## E.1  Text-based Policy

### E.1.1  Skill Discovery Prompt

---

**ALFWorld: Skill discovery prompt**

You will be given the interaction histories between an agent and a household
environment. Break the full interaction log into several sub-tasks, and
summarize these sub-tasks in an abstract way. When there is already a list of
environment-specific subtasks to start with, please keep them in your response
or improve them if you figure out any mistakes, and you can also append
new subtasks according to the given new trial. You should not add duplicate
subtasks. Give your response after "New summarization: "

Here is an example:

Task 1:
Success trial:
You are in the middle of a room. Looking quickly around you, you see a cabinet
4, a cabinet 3, a cabinet 2, a cabinet 1, a coffeemachine 1, a countertop 1,
a diningtable 3, a diningtable 2, a diningtable 1, a drawer 1, a fridge 1, a
garbagecan 1, a microwave 1, a sidetable 1, a sinkbasin 1, a stoveburner 4, a
stoveburner 3, a stoveburner 2, a stoveburner 1, and a toaster 1.
Your task is to: find some apple and put it in sidetable.
> go to fridge 1
The fridge 1 is closed.
> open fridge 1
You open the fridge 1. The fridge 1 is open. In it, you see a lettuce 2, a mug
2, a potato 2, and a tomato 1.
> close fridge 1
> go to garbagecan 1
On the garbagecan 1, you see a apple 3, and a egg 3.
> take apple 3 from garbagecan 1
You pick up the apple 3 from the garbagecan 1.
> go to sidetable 1
On the sidetable 1, you see a cup 1, a lettuce 1, a peppershaker 3, a potato 1,
and a saltshaker 1.
> put apple 3 in/on sidetable 1
You put the apple 3 in/on the sidetable 1.

Here is no list of subtasks to start with.

New summarization:

In the above interaction log, the agent completes three subtasks: find(apple)
including "go to", "open" and "close" as primitive operations. take(apple)
including "take" as primitive operations, and put(apple, sidetable) including
"go to" and "put" as primitive operations. We thus can segment the entire log
based on these three subtasks below.

You are in the middle of a room. Looking quickly around you, you see a cabinet
4, a cabinet 3, a cabinet 2, a cabinet 1, a coffeemachine 1, a countertop 1,
a diningtable 3, a diningtable 2, a diningtable 1, a drawer 1, a fridge 1, a
garbagecan 1, a microwave 1, a sidetable 1, a sinkbasin 1, a stoveburner 4, a
stoveburner 3, a stoveburner 2, a stoveburner 1, and a toaster 1.
Your task is to: find some apple and put it in sidetable.
- find(apple):
    > go to fridge 1
    The fridge 1 is closed.
    > open fridge 1

---

```
    You open the fridge 1. The fridge 1 is open. In it, you see a lettuce 2, a
mug 2, a potato 2, and a tomato 1.
    > close fridge 1
    You close fridge 1.
    > go to garbagecan 1
    On the garbagecan 1, you see a apple 3, and a egg 3.
- take(apple):
    > take apple 3 from garbagecan 1
    You pick up the apple 3 from the garbagecan 1.
- put(apple, sidetable):
    > go to sidetable 1
    On the sidetable 1, you see a cup 1, a lettuce 1, a peppershaker 3, a
potato 1, and a saltshaker 1.
    > put apple 3 in/on sidetable 1
    You put the apple 3 in/on the sidetable 1.

As there is no a list of subtasks to start with, we list the
Enviroment-specific subtasks below:

- find(object)
- take(object)
- put(object, receptable)
```

**WebShop: Skill discovery prompt**

```
You will be given the interaction histories between an agent and a webshop
environment. Break the full interaction log into several sub-tasks, and
summarize these sub-tasks in an abstrat way. When there is already a list of
environment-specific subtasks to start with, please keep them in your response
or improve them if you figure out any mistakes, and you can also append
new subtasks according to the given new trial. You should not add duplicate
subtasks. Give your response after "New summarization: "

Here is an example:

Task 1:
Success trial:
WebShop
Instruction:
i'm looking for a light pink long handle back loofah shower brush, and price
lower than 40.00 dollars
[Search]

Action: search[light pink long handle back loofah shower brush]
Observation:
[Back to Search]
Page 1 (Total results: 50)
[Next >]
[B09N8YLMGJ]
JulaJuyo Shower Back Scrubber, Long Handle Bath Body Brush Exfoliating Loofah
Sponge on a Stick for Men Women, Colorful Nylon Mesh Showering Bathing Luffa
Cleaning Lufa Washer Brush (1 Pack-Gray)
\$7.99
[B09J8KGN4P]
JulaJuyo Back Scrubber for Shower, Long Handle Body Bath Brush Showering Loofah
Sponge on a Stick for Men Women, Nylon Mesh Exfoliating Bathing Lufa Scrub
Shower Cleaning Luffa Brush (1 Pack-White)
\$7.99
[B08Y6VZFT6]
Prozklves Back Scrubber for Shower, Long Handle Back Loofah Shower Brush, Soft
Nylon Mesh Back Cleaner Washer, Bath Brush for Women Men, Exfoliating Body
Scrubber for Elderly (Light Pink)
\$7.99
```

```
Action: click[B08Y6VZFT6]
Observation:
[Back to Search]
[< Prev]
scent [2 pack-beige][2 pack-green][2 pack-light pink][2
pcs=green+beige][2 pcs=green+pink][3 pack-beige][3 pack-light pink][3
pcs=colors][beige][green][light pink]
Prozklves Back Scrubber for Shower, Long Handle Back Loofah Shower Brush, Soft
Nylon Mesh Back Cleaner Washer, Bath Brush for Women Men, Exfoliating Body
Scrubber for Elderly (Light Pink)
Price:\$7.99
Rating: N.A.
[Description]
[Features]
[Reviews]
[Attributes]
[Buy Now]

Here is no list of subtasks to start with.

New summarization:

In the above interaction log, the agent completes two subtasks: search_item()
with the purpose of searching for the required item, including "search[QUERY]"
as primitive operation, as well as select_item() with the purpose of selecting
the item that matches the requirement, including "click[ITEM_TITLE]" as a
primitive operation. We thus can segment the entire log based on these two
subtasks below.

WebShop
Instruction:
i'm looking for a light pink long handle back loofah shower brush, and price
lower than 40.00 dollars
[Search]

- search_item():
    Action: search[light pink long handle back loofah shower brush]
    Observation:
    [Back to Search]
    Page 1 (Total results: 50)
    [Next >]
    [B09N8YLMGJ]
    JulaJuyo Shower Back Scrubber, Long Handle Bath Body Brush Exfoliating
Loofah Sponge on a Stick for Men Women, Colorful Nylon Mesh Showering Bathing
Luffa Cleaning Lufa Washer Brush (1 Pack-Gray)
    \$7.99
    [B09J8KGN4P]
    JulaJuyo Back Scrubber for Shower, Long Handle Body Bath Brush Showering
Loofah Sponge on a Stick for Men Women, Nylon Mesh Exfoliating Bathing Lufa
Scrub Shower Cleaning Luffa Brush (1 Pack-White)
    \$7.99
    [B08Y6VZFT6]
    Prozklves Back Scrubber for Shower, Long Handle Back Loofah Shower Brush,
Soft Nylon Mesh Back Cleaner Washer, Bath Brush for Women Men, Exfoliating Body
Scrubber for Elderly (Light Pink)
    \$7.99
- select_item():
    Action: click[B08Y6VZFT6]
    Observation:
    [Back to Search]
    [< Prev]
```

```
    scent [2 pack-beige][2 pack-green][2 pack-light pink][2
pcs=green+beige][2 pcs=green+pink][3 pack-beige][3 pack-light pink][3
pcs=colors][beige][green][light pink]
    Prozklves Back Scrubber for Shower, Long Handle Back Loofah Shower Brush,
Soft Nylon Mesh Back Cleaner Washer, Bath Brush for Women Men, Exfoliating Body
Scrubber for Elderly (Light Pink)
    Price: \$7.99
    Rating: N.A.
    [Description]
    [Features]
    [Reviews]
    [Attributes]
    [Buy Now]

As there is no a list of subtasks to start with, we list the
Enviroment-specific subtasks below:

- search_item()
- select_item()
```

### E.1.2 Primitive Discovery Prompt

**ALFWorld: Skill-conditioned primitive discovery prompt**

```
You will be given success histories in which you were placed an environment and
given a task to complete. Please extract the valid environment-specific actions
with the correct syntax from the trial, and list them under the corresponding
subtask catergory. When there is already a list of environment-specific actions
to start with, please keep them in your response or improve them if any syntax
error detected, and you can also append new actions according to the given new
trial. You should not add duplicate actions regardless of the examples. For
existing actions in the list, do not add more examples. Give your response
after "New summarization: "

Here is one example:

Task 1:
Success trial:
Plan for Completing the Task:
1. find(apple)
2. take(apple)

Segmented Interaction History:
> > find(apple): find some apple
> go to garbagecan 1
On the garbagecan 1, you see a apple 3, and a egg 3.
> > take(apple): take the apple
> take apple 3 from garbagecan 1
You pick up the apple 3 from the garbagecan 1.

New summarization:

> > find(object)
Environment-specific actions:
    - go to [receptacle] [id]
    Example: go to garbagecan 1

> > take(object)
Environment-specific actions:
    - take [object] [id] from [receptacle] [id]
    Example: take apple 3 from garbagecan 1

> > put(object, receptacle)
```

```
Environment-specific actions:

> > cool(object)
Environment-specific actions:

> > use(object)
Environment-specific actions:

> > clean(object)
Environment-specific actions:

> > heat(object)
Environment-specific actions:
```

**WebShop: Skill-conditioned primitive discovery prompt**

```
You will be given success histories in which you were placed an environment and
given a task to complete. Please extract the valid environment-specific actions
with the correct syntax from the trial, and list them under the corresponding
subtask catergory. When there is already a list of environment-specific actions
to start with, please keep them in your response or improve them if any syntax
error detected, and you can also append new actions according to the given new
trial. You should not add duplicate actions regardless of the examples. For
existing actions in the list, do not add more examples. Give your response
after "New summarization: "

Here is one example:

Webshop
Instruction:
i would like a 3 ounce bottle of bright citrus deodorant for sensitive skin,
and price lower than 50.00 dollars
[Search]

> > search_item(): search with a query regarding the instruction
Action: search[3 ounce bright citrus deodorant sensitive skin]
Observation:
[Back to Search]
Page 1 (Total results: 50)
[Next >]
[B078GWRC1J]
Bright Citrus Deodorant by Earth Mama | Natural and Safe for Sensitive Skin,
Pregnancy and Breastfeeding, Contains Organic Calendula 3-Ounce
\$10.99
[B078GTKVXY]
Ginger Fresh Deodorant by Earth Mama | Natural and Safe for Sensitive Skin,
Pregnancy and Breastfeeding, Contains Organic Calendula 3-Ounce
\$10.99
[B08KBVJ4XN]
Barrel and Oak - Aluminum-Free Deodorant, Deodorant for Men, Essential
Oil-Based Scent, 24-Hour Odor Protection, Cedar & Patchouli Blend, Gentle on
Sensitive Skin (Mountain Sage, 2.7 oz, 2-Pack)
\$15.95

Action: click[Back to Search]
Observation:
Webshop
Instruction:
i would like a 3 ounce bottle of bright citrus deodorant for sensitive skin,
and price lower than 50.00 dollars
[Search]
```

```
New summarization:

> > search_item()
Environment-specific actions:
    search[Query]
    # Example search[3 ounce bright citrus deodorant sensitive skin]
    click[Back to Search]
    # Example click[Back to Search]

> > select_item()
Environment-specific actions:

> > select_item_attributes()
Environment-specific actions:

> > purchase_item()
Environment-specific actions:
```

### E.1.3    Skill-conditioned Policy Improvement Tips Distillation Prompt

**ALFWorld: Policy improvement tips contrastive distillation prompt**

```
You will be given both failure and success histories in which you were placed
in an environment and given a task to complete. You should generate tips
according to your task description rather than the historical interactions.
By comparing the two histories, please find the mistakes in the failure trial,
and summarize tips to fix the mistakes. The tips can also be the rules in this
particular environment. You will need these tips later to solve a similar task.
When there is already a list of tips to start with, please keep them in your
response or improve them, and you can also append new tips according to the
given new trials. If the new mistake you detect can be solved by any existing
tips, you do not need to generate new tips. You do not have to add new tips if
there is no one. Give your response after "New summarization: "

Here is one example:

Task 1:
Failure trial:
You are in the middle of a room. Looking quickly around you, you see a cabinet
10, a cabinet 9, a cabinet 8, a cabinet 7, a cabinet 6, a cabinet 5, a cabinet
4, a cabinet 3, a cabinet 2, a cabinet 1, a coffeemachine 1, a countertop
2, a countertop 1, a diningtable 2, a diningtable 1, a drawer 3, a drawer
2, a drawer 1, a fridge 1, a garbagecan 1, a microwave 1, a sinkbasin 1, a
stoveburner 4, a stoveburner 3, a stoveburner 2, a stoveburner 1, and a toaster
1.
Your historical interactions:
> go to countertop 1
On the countertop 1, you see a apple 2, a butterknife 1, a cup 1, a knife 2, a
knife 1, a pan 1, a potato 2, a spatula 2, a spatula 1, a spoon 2, a spoon 1, a
tomato 2, a winebottle 3, and a winebottle 1.
> go to countertop 2
On the countertop 2, you see a butterknife 1, a cup 1, a fork 2, and a mug 1.
> take mug 1 from countertop 2
You pick up the mug 1 from the countertop 2.
Your task is to: heat the mug
> go to stove burner 1
On the stoveburner 1, you see a pan 2.
> heat mug 1 with stoveburner 1
Nothing happens.
> heat mug 1 with stoveburner 1
Nothing happens.

Success trial:
```

```
You are in the middle of a room. Looking quickly around you, you see a cabinet
10, a cabinet 9, a cabinet 8, a cabinet 7, a cabinet 6, a cabinet 5, a cabinet
4, a cabinet 3, a cabinet 2, a cabinet 1, a coffeemachine 1, a countertop
2, a countertop 1, a diningtable 2, a diningtable 1, a drawer 3, a drawer
2, a drawer 1, a fridge 1, a garbagecan 1, a microwave 1, a sinkbasin 1, a
stoveburner 4, a stoveburner 3, a stoveburner 2, a stoveburner 1, and a toaster
1.
Your historical interactions:
> go to countertop 1
On the countertop 1, you see a apple 2, a butterknife 1, a cup 1, a knife 2, a
knife 1, a pan 1, a potato 2, a spatula 2, a spatula 1, a spoon 2, a spoon 1, a
tomato 2, a winebottle 3, and a winebottle 1.
> go to countertop 2
On the countertop 2, you see a butterknife 1, a cup 1, a fork 2, and a mug 1.
> take mug 1 from countertop 2
You pick up the mug 1 from the countertop 2.
Your task is to: heat the mug
> go to microwave 1
You arrive at loc 22. The microwave 1 is closed.
> heat mug 1 with microwave 1
You heat the mug 1 using the microwave 1.

New summarization: In task 1, my task is to heat the mug 1. I made a mistake
in heating the mug with a stoveburner, and I also got stuck in a loop where I
continually did this. As shown in the success trial, the correct way is to heat
a mug with a microwave. Thus, I should try different actions when getting stuck
in a loop; I should use a microwave to heat a mug, and I can directly heat an
object with a microwave without opening the microwave and putting the object in
it.

Tips:
- Try to execute a different action when you get stuck in a loop of taking
identical actions.
- Mug should be heated by a microwave rather than a stoveburner.
- You can directly heat an object with a microwave by taking the corresponding
action.

Note, you should not directly include the above example tips in your response.
Give your response after "New summarization: "
```

**ALFWorld: Policy improvement tips non-contrastive distillation prompt**

```
You will be given success histories in which you were placed in an environment
and given a task to complete. Please summarize tips according to your task
description rather than the historical interactions.The tips can be the rules
in this particular environment. You will need these tips later to solve a
similar task. When there is already a list of tips to start with, please
keep them in your response or improve them, and you can also append new tips
according to the given new trials. You do not have to add new tips if there is
no one. Give your response after "New summarization: "

Here is one example:

Task 1:
Success trial:
You are in the middle of a room. Looking quickly around you, you see a cabinet
10, a cabinet 9, a cabinet 8, a cabinet 7, a cabinet 6, a cabinet 5, a cabinet
4, a cabinet 3, a cabinet 2, a cabinet 1, a coffeemachine 1, a countertop
2, a countertop 1, a diningtable 2, a diningtable 1, a drawer 3, a drawer
2, a drawer 1, a fridge 1, a garbagecan 1, a microwave 1, a sinkbasin 1, a
stoveburner 4, a stoveburner 3, a stoveburner 2, a stoveburner 1, and a toaster
1.
Your historical interactions:
```

```
> go to countertop 1
On the countertop 1, you see a apple 2, a butterknife 1, a cup 1, a knife 2, a
knife 1, a pan 1, a potato 2, a spatula 2, a spatula 1, a spoon 2, a spoon 1, a
tomato 2, a winebottle 3, and a winebottle 1.
> go to countertop 2
On the countertop 2, you see a butterknife 1, a cup 1, a fork 2, and a mug 1.
> take mug 1 from countertop 2
You pick up the mug 1 from the countertop 2.
Your task is to: heat the mug
> go to microwave 1
You arrive at loc 22. The microwave 1 is closed.
> heat mug 1 with microwave 1
You heat the mug 1 using the microwave 1.

New summarization: In task 1, my task is to heat the mug 1. As shown in the
success trial, the correct way is to heat a mug with a microwave. Thus, I
should use a microwave to heat a mug, and I can directly heat an object with
a microwave without opening the microwave and putting the object in it.

Tips:
- Mug should be heated by a microwave.
- You can directly heat an object with a microwave by taking the corresponding
action.

Note, you should not directly include the above example tips in your response.
Give your response after "New summarization: "
```

**WebShop: Policy improvement tips contrastive distillation prompt**

```
You will be given both failure and success histories in which you were placed
in an environment and given a task to complete. You should generate tips
according to your task description rather than the historical interactions.
By comparing the two histories, please find the mistakes in the failure trial,
and summarize general tips to fix the mistakes. The tips can also be about the
rules in this particular environment. You will need these tips later to solve
a similar task. When there is already a list of tips to start with, please
keep them in your response or improve them, and you can also append new tips
according to the given new trials. If the new mistake you detect can be solved
by any existing tips, you do not need to generate new tips. You do not have to
add new tips if there is no one. Give your response after "New summarization: "

Here is one example:

Task1:
Failure Trial:
Webshop
Instruction:
i would like a 3 ounce bottle of bright citrus deodorant for sensitive skin,
and price lower than 50.00 dollars
[Search]
Your historical interactions:
Action: search[3 ounce bright citrus deodorant sensitive skin]
Observation:
[Back to Search]
Page 1 (Total results: 50)
[Next >]
[B078GWRC1J]
Bright Citrus Deodorant by Earth Mama | Natural and Safe for Sensitive Skin,
Pregnancy and Breastfeeding, Contains Organic Calendula 3-Ounce
\$10.99
[B078GTKVXY]
Ginger Fresh Deodorant by Earth Mama | Natural and Safe for Sensitive Skin,
Pregnancy and Breastfeeding, Contains Organic Calendula 3-Ounce
```

```
\$10.99
[B08KBVJ4XN]
Barrel and Oak - Aluminum-Free Deodorant, Deodorant for Men, Essential
Oil-Based Scent, 24-Hour Odor Protection, Cedar & Patchouli Blend, Gentle on
Sensitive Skin (Mountain Sage, 2.7 oz, 2-Pack)
\$15.95
Action: click[B078GWRC1J]
Observation:
[Back to Search]
[< Prev]
scent [assorted scents][bright citrus][calming lavender][ginger fresh][simply
non-scents]
size [travel set (4-pack)][3 ounce (pack of 1)][3-ounce (2-pack)]
Bright Citrus Deodorant by Earth Mama | Natural and Safe for Sensitive Skin,
Pregnancy and Breastfeeding, Contains Organic Calendula 3-Ounce
Price: \$10.99
Rating: N.A.
[Description]
[Features]
[Reviews]
[Buy Now]
Your task is: select_item_attributes()
Action: click[BRIGHT CITRUS]
Observation: Invalid action!

Success Trial:
Webshop
Instruction:
i would like a 3 ounce bottle of bright citrus deodorant for sensitive skin,
and price lower than 50.00 dollars
[Search]
Your historical interactions:
Action: search[3 ounce bright citrus deodorant sensitive skin]
Observation:
[Back to Search]
Page 1 (Total results: 50)
[Next >]
[B078GWRC1J]
Bright Citrus Deodorant by Earth Mama | Natural and Safe for Sensitive Skin,
Pregnancy and Breastfeeding, Contains Organic Calendula 3-Ounce
\$10.99
[B078GTKVXY]
Ginger Fresh Deodorant by Earth Mama | Natural and Safe for Sensitive Skin,
Pregnancy and Breastfeeding, Contains Organic Calendula 3-Ounce
\$10.99
[B08KBVJ4XN]
Barrel and Oak - Aluminum-Free Deodorant, Deodorant for Men, Essential
Oil-Based Scent, 24-Hour Odor Protection, Cedar & Patchouli Blend, Gentle on
Sensitive Skin (Mountain Sage, 2.7 oz, 2-Pack)
\$15.95
Action: click[B078GWRC1J]
Observation:
[Back to Search]
[< Prev]
scent [assorted scents][bright citrus][calming lavender][ginger fresh][simply
non-scents]
size [travel set (4-pack)][3 ounce (pack of 1)][3-ounce (2-pack)]
Bright Citrus Deodorant by Earth Mama | Natural and Safe for Sensitive Skin,
Pregnancy and Breastfeeding, Contains Organic Calendula 3-Ounce
Price: \$10.99
Rating: N.A.
[Description]
```

```
[Features]
[Reviews]
[Buy Now]
Your task is: select_item_attributes()
Action: click[bright citrus]
Observation: You have clicked bright citrus.
Action: click[3 ounce (pack of 1)]
Observation: You have clicked 3 ounce (pack of 1).

New summarization: In task 1, my task is select_item_attributes(), I made a
mistake on capitalizing the name of a scent option [bright citrus], so that
click[BRIGHT CITRUS] become invalid. As shown in the success trial, the correct
action should be click[bright citrus]. Thus, I should strictly use the exact
words of each option without making any changes.

Tips:
- You should strictly use the exact words of each available option without
making any changes.

Note, do not directly include the above example tips in your response. Give
your response after "New summarization: "
```

**WebShop: Policy improvement tips non-contrastive distillation prompt**

```
You will be given success histories in which you were placed in an environment
and given a task to complete. Please summarize tips according to your task
description rather than the historical interactions. The tips can be the
rules in this particular environment. You will need these tips later to solve
a similar task. When there is already a list of tips to start with, please
keep them in your response or improve them, and you can also append new tips
according to the given new trials. You do not have to add new tips if there is
no one. Give your response after "New summarization: "

Here is one example:

Task1:
Success Trial:
Webshop
Instruction:
i would like a 3 ounce bottle of bright citrus deodorant for sensitive skin,
and price lower than 50.00 dollars
[Search]
Your historical interactions:
Action: search[3 ounce bright citrus deodorant sensitive skin]
Observation:
[Back to Search]
Page 1 (Total results: 50)
[Next >]
[B078GWRC1J]
Bright Citrus Deodorant by Earth Mama | Natural and Safe for Sensitive Skin,
Pregnancy and Breastfeeding, Contains Organic Calendula 3-Ounce
\$10.99
[B078GTKVXY]
Ginger Fresh Deodorant by Earth Mama | Natural and Safe for Sensitive Skin,
Pregnancy and Breastfeeding, Contains Organic Calendula 3-Ounce
\$10.99
[B08KBVJ4XN]
Barrel and Oak - Aluminum-Free Deodorant, Deodorant for Men, Essential
Oil-Based Scent, 24-Hour Odor Protection, Cedar & Patchouli Blend, Gentle on
Sensitive Skin (Mountain Sage, 2.7 oz, 2-Pack)
\$15.95
Action: click[B078GWRC1J]
```

```
Observation:
[Back to Search]
[< Prev]
scent [assorted scents][bright citrus][calming lavender][ginger fresh][simply
non-scents]
size [travel set (4-pack)][3 ounce (pack of 1)][3-ounce (2-pack)]
Bright Citrus Deodorant by Earth Mama | Natural and Safe for Sensitive Skin,
Pregnancy and Breastfeeding, Contains Organic Calendula 3-Ounce
Price: \$10.99
Rating: N.A.
[Description]
[Features]
[Reviews]
[Buy Now]
Your task is: select_item_attributes()
Action: click[bright citrus]
Observation: You have clicked bright citrus.
Action: click[3 ounce (pack of 1)]
Observation: You have clicked 3 ounce (pack of 1).

New summarization: In task 1, my task is select_item_attributes(). As shown in
the success trial, I selected attributes according to the instruction.

Tips:
- You should selected attributes according to instruction.

Note, do not directly include the tips in the above example in your response.
Give your response after "New summarization: "
```

### E.1.4   Prompts for Downstream Task Solving

**ALFWorld: Base policy prompt**

```
Interact with a household to complete a given task.
You can call subprocedures find(object), take(object), use(object),
clean(object), heat(object), cool(object) and put(object, location). Here are
some examples.

You are in the middle of a room. Looking quickly around you, you see a cabinet
4, a cabinet 3, a cabinet 2, a cabinet 1, a countertop 1, a garbagecan 1, a
handtowelholder 2, a handtowelholder 1, a sinkbasin 2, a sinkbasin 1, a toilet
1, a toiletpaperhanger 1, and a towelholder 1.
Your task is to: put some spraybottle on toilet.
> think: To solve the task, I need to find and take a sparybottle, then put it
on toilet.
OK.
> call: find(spraybottle)
Subprocedure finished.
> call: take(spraybottle)
Subprocedure finished.
> call: put(spraybottle, toilet)
Subprocedure finished.

You are in the middle of a room. Looking quickly around you, you see a cabinet
4, a cabinet 3, a cabinet 2, a cabinet 1, a coffeemachine 1, a countertop 1,
a diningtable 3, a diningtable 2, a diningtable 1, a drawer 1, a fridge 1, a
garbagecan 1, a microwave 1, a sidetable 1, a sinkbasin 1, a stoveburner 4, a
stoveburner 3, a stoveburner 2, a stoveburner 1, and a toaster 1.
Your task is to: find some apple and put it in sidetable.
> think: To solve the task, I need to find and take an apple, then put it in
sidetable.
OK.
> call: find(apple)
```

```
Subprocedure finished.
> call: take(apple)
Subprocedure finished.
> call: put(apple, sidetable)
Subprocedure finished.

You are in the middle of a room. Looking quickly around you, you see a cabinet
13, a cabinet 12, a cabinet 11, a cabinet 10, a cabinet 9, a cabinet 8, a
cabinet 7, a cabinet 6, a cabinet 5, a cabinet 4, a cabinet 3, a cabinet 2,
a cabinet 1, a coffeemachine 1, a countertop 1, a diningtable 1, a drawer 1,
a fridge 1, a garbagecan 1, a microwave 1, a shelf 3, a shelf 2, a shelf 1, a
sinkbasin 1, a stoveburner 4, a stoveburner 3, a stoveburner 2, a stoveburner 1,
and a toaster 1.
Your task is to: put a clean lettuce in diningtable.
> think: To solve the task, I need to find and take a lettuce, then clean it
with sinkbasin, then put it in diningtable.
OK.
> call: find(lettuce)
Subprocedure finished.
> call: take(lettuce)
Subprocedure finished.
> call: clean(lettuce)
Subprocedure finished.
> call: put(lettuce, diningtable)
Subprocedure finished.

You are in the middle of a room. Looking quickly around you, you see a cabinet
10, a cabinet 9, a cabinet 8, a cabinet 7, a cabinet 6, a cabinet 5, a cabinet
4, a cabinet 3, a cabinet 2, a cabinet 1, a coffeemachine 1, a countertop 3,
a countertop 2, a countertop 1, a diningtable 1, a drawer 6, a drawer 5, a
drawer 4, a drawer 3, a drawer 2, a drawer 1, a fridge 1, a garbagecan 1, a
microwave 1, a sinkbasin 1, a stoveburner 4, a stoveburner 3, a stoveburner 2,
a stoveburner 1, and a toaster 1.
Your task is to: heat some egg and put it in diningtable.
> think: To solve the task, I need to find and take an egg, then heat it with
microwave, then put it in diningtable.
OK.
> call: find(egg)
Subprocedure finished.
> call: take(egg)
Subprocedure finished.
> call: heat(egg)
Subprocedure finished.
> call: put(egg, diningtable)
Subprocedure finished.

You are in the middle of a room. Looking quickly around you, you see a cabinet
13, a cabinet 12, a cabinet 11, a cabinet 10, a cabinet 9, a cabinet 8, a
cabinet 7, a cabinet 6, a cabinet 5, a cabinet 4, a cabinet 3, a cabinet 2,
a cabinet 1, a coffeemachine 1, a countertop 1, a diningtable 1, a drawer 1,
a fridge 1, a garbagecan 1, a microwave 1, a shelf 3, a shelf 2, a shelf 1, a
sinkbasin 1, a stoveburner 4, a stoveburner 3, a stoveburner 2, a stoveburner 1,
and a toaster 1.
Your task is to: put a hot apple in fridge.
> think: To solve the task, I need to find and take an apple, then heat it with
microwave, then put it in fridge.
OK.
> call: find(apple)
Subprocedure finished.
> call: take(apple)
Subprocedure finished.
> call: heat(apple)
Subprocedure finished.
```

```
> call: put(apple, fridge)
Subprocedure finished.

You are in the middle of a room. Looking quickly around you, you see a cabinet
16, a cabinet 15, a cabinet 14, a cabinet 13, a cabinet 12, a cabinet 11, a
cabinet 10, a cabinet 9, a cabinet 8, a cabinet 7, a cabinet 6, a cabinet
5, a cabinet 4, a cabinet 3, a cabinet 2, a cabinet 1, a coffeemachine 1, a
countertop 2, a countertop 1, a diningtable 1, a drawer 5, a drawer 4, a drawer
3, a drawer 2, a drawer 1, a fridge 1, a garbagecan 1, a microwave 1, a safe 1,
a sinkbasin 1, a stoveburner 4, a stoveburner 3, a stoveburner 2, a stoveburner
1, and a toaster 1.
Your task is to: cool some pan and put it in stoveburner.
> think: To solve the task, I need to find and take a pan, then cool it with
fridge, then put it in stoveburner.
OK.
> call: find(pan)
Subprocedure finished.
> call: take(pan)
Subprocedure finished.
> call: cool(pan)
Subprocedure finished.
> call: put(pan, stoveburner)
Subprocedure finished.

You are in the middle of a room. Looking quickly around you, you see a
coffeetable 1, a diningtable 1, a drawer 4, a drawer 3, a drawer 2, a drawer
1, a dresser 1, a garbagecan 1, a sidetable 2, a sidetable 1, and a sofa 1.
Your task is to: put two cellphone in sofa.
> think: To solve the task, I need to find and take the first cellphone, then
put it in sofa, then find and take the second cellphone, then put it in sofa.
OK.
> call: find(cellphone1)
Subprocedure finished.
> call: take(cellphone1)
Subprocedure finished.
> call: put(cellphone1, sofa)
Subprocedure finished.
> call: find(cellphone2)
Subprocedure finished.
> call: take(cellphone2)
Subprocedure finished.
> call: put(cellphone2, sofa)
Subprocedure finished.

You are in the middle of a room. Looking quickly around you, you see a bed 1, a
desk 1, a drawer 3, a drawer 2, a drawer 1, a garbagecan 1, a safe 1, a shelf 5,
a shelf 4, a shelf 3, a shelf 2, a shelf 1, a sidetable 2, and a sidetable 1.
Your task is to: look at bowl under the desklamp.
> think: To solve the task, I need to find and take a bowl, then find and use a
desklamp.
OK.
> call: find(bowl)
Subprocedure finished.
> call: take(bowl)
Subprocedure finished.
> call: find(desklamp)
Subprocedure finished.
> call: use(desklamp)
Subprocedure finished.

You are in the middle of a room. Looking quickly around you, you see a bed 1, a
drawer 10, a drawer 9, a drawer 8, a drawer 7, a drawer 6, a drawer 5, a drawer
4, a drawer 3, a drawer 2, a drawer 1, a dresser 1, a garbagecan 1, a shelf 9,
```

```
a shelf 8, a shelf 7, a shelf 6, a shelf 5, a shelf 4, a shelf 3, a shelf 2,
and a shelf 1.
Your task is to: examine the pen with the desklamp.
> think: To solve the task, I need to find and take a pen, then find and use a
desklamp. First I need to find a pen. A pen is more likely to appear in drawer
(1-10), shelf (1-9), bed (1), garbagecan (1). I can check one by one, starting
with drawer 1.
OK.
> call: find(pen)
Subprocedure finished.
> call: take(pen)
Subprocedure finished.
> call: find(desklamp)
Subprocedure finished.
> call: use(desklamp)
Subprocedure finished.
```

There are 7 skills in ALFWorld, which are find(object), take(object), use(object), clean(object), heat(object), cool(object) and put(object, location). Each of the skill induces a skill-conditioned policy. Here we first give a general prompt template for these policies.

---

**ALFWorld: skill-conditioned policy prompt template**

```
Interact with a household to complete a given task. Please consider the
following tips to solve new tasks:

‘‘‘
{policy improvement tips}
{primitives}
‘‘‘

Here are some examples.

‘‘‘
{examples}
‘‘‘
```

---

In the above template, the policy improvement tips and primitives for each skill are provided in Appendix E.4. For the examples, we use the same examples provided by ReAct. Note that ReAct uses different examples to prompt different types of ALFWorld tasks, while we reorganize the examples such that a single set of prompts is used across all types of tasks. This is because we discovered skills that are shared across tasks, allowing more generalizable knowledge.

Below we provide the full prompt of a skill-conditioned policy for skill "heat(object)". Prompts of other skills are available in our codebased, which will be released soon.

---

**ALFWorld: skill-conditioned policy prompt for heat(object)**

```
Interact with a household to complete a given task. Please consider the
following rules to solve new tasks:
1. Try to execute a different action when you get stuck in a loop of taking
identical actions.
2. Ensure you have the correct object in your possession before attempting to
perform actions on it.
3. Check all possible locations if you can't find an object in the expected
places.
4. You can directly heat an object with a microwave by taking the corresponding
action.
5. If an action does not work, try a different approach or command.
6. Focus on relevant locations to complete the task efficiently.
7. A cup is not the same as a mug. Make sure to find and use the correct object
for the task.
```

---

```
8. You can think about your plan in the first step, and after that you can only
select from the following actions in your response (please strictly follow the
syntax):
    go to [receptacle] [id]
    # Example:
        go to countertop 1
    heat [object] [id] with [receptacle] [id]
    # Example:
        heat egg 1 with microwave 1
```

Here are some examples.

You are in the middle of a room. Looking quickly around you, you see a cabinet
10, a cabinet 9, a cabinet 8, a cabinet 7, a cabinet 6, a cabinet 5, a cabinet
4, a cabinet 3, a cabinet 2, a cabinet 1, a coffeemachine 1, a countertop 3,
a countertop 2, a countertop 1, a diningtable 1, a drawer 6, a drawer 5, a
drawer 4, a drawer 3, a drawer 2, a drawer 1, a fridge 1, a garbagecan 1, a
microwave 1, a sinkbasin 1, a stoveburner 4, a stoveburner 3, a stoveburner 2,
a stoveburner 1, and a toaster 1.
Your historical interactions:
> open fridge 1
You open the fridge 1. The fridge 1 is open. In it, you see a lettuce 2, a mug
2, and a potato 3.
> go to countertop 1
On the countertop 1, you see a bread 1, a fork 1, and a saltshaker 1.
> go to countertop 2
On the countertop 2, you see nothing.
> go to countertop 3
On the countertop 3, you see a bowl 1, a butterknife 1, a egg 2, a kettle 2, a
plate 1, a sink 1, and a spatula 2.
> take egg 2 from countertop 3
You pick up the egg 2 from the countertop 3.
Your task is: heat(egg)
> think: My task is to heat an egg. Now I take an egg (2). Next, I need go to a
microwave (1) and heat it.
OK.
> go to microwave 1
The microwave 1 is closed.
> heat egg 2 with microwave 1
You heat the egg 2 using the microwave 1.
> Done heat(egg): I have heated the desired egg (2).

You are in the middle of a room. Looking quickly around you, you see a cabinet
13, a cabinet 12, a cabinet 11, a cabinet 10, a cabinet 9, a cabinet 8, a
cabinet 7, a cabinet 6, a cabinet 5, a cabinet 4, a cabinet 3, a cabinet 2,
a cabinet 1, a coffeemachine 1, a countertop 1, a diningtable 1, a drawer 1,
a fridge 1, a garbagecan 1, a microwave 1, a shelf 3, a shelf 2, a shelf 1, a
sinkbasin 1, a stoveburner 4, a stoveburner 3, a stoveburner 2, a stoveburner 1,
and a toaster 1.
Your historical interactions:
> go to fridge 1
The fridge 1 is closed.
> open fridge 1
You open the fridge 1. The fridge 1 is open. In it, you see a cup 1, and a egg
1.
> go to diningtable 1
On the diningtable 1, you see a apple 1, a bread 1, a fork 3, a fork 2, a fork
1, a mug 2, a peppershaker 3, a plate 2, a pot 1, a soapbottle 3, a spatula 1,
a spoon 3, a spoon 2, a spoon 1, a winebottle 3, and a winebottle 2.
> take apple 1 from diningtable 1
You pick up the apple 1 from the diningtable 1.
Your task is: heat(apple)
```

```
> think: My task is to heat an apple. Now I take an apple (1). Next, I need to
go to a microwave (1) and heat it.
OK.
> go to microwave 1
The microwave 1 is closed.
> heat apple 1 with microwave 1
You heat the apple 1 using the microwave 1.
> Done heat(apple): I have heated the desired apple (1).
'''
```

We note that in WebShop, whether to call a skill depends on the environment feedback. Therefore, the base policy needs to act based on environment observation at the end of each skill. For example, if the current item has no available attribute to select, then the base policy will not call the *select_item_attributes()* skill. Otherwise it will first select attributes then purchase the item. Below is the prompt for the base policy.

---

**WebShop: Base policy prompt**

```
Interact with a webshop to solve a task.
You can call subprocedures search_item(), select_item(),
select_item_attributes(), purchase_item(). Here are some examples.

Webshop
Instruction:
i would like a 3 ounce bottle of bright citrus deodorant for sensitive skin,
and price lower than 50.00 dollars
[Search]

Action: think[I need to first search for available items, then select the item
that matches the instruction. If there are additional options for certain
attributes, I also need to select them before purchase. Otherwise I can
directly purchase the item.]
Observation: OK.

Action: search_item()
Observation:
Subprocedure Done[I have found a list of items.]
Current Page:
[Back to Search]
Page 1 (Total results: 50)
[Next >]
[B078GWRC1J]
Bright Citrus Deodorant by Earth Mama | Natural and Safe for Sensitive Skin,
Pregnancy and Breastfeeding, Contains Organic Calendula 3-Ounce
\$10.99
[B078GTKVXY]
Ginger Fresh Deodorant by Earth Mama | Natural and Safe for Sensitive Skin,
Pregnancy and Breastfeeding, Contains Organic Calendula 3-Ounce
\$10.99
[B08KBVJ4XN]
Barrel and Oak - Aluminum-Free Deodorant, Deodorant for Men, Essential
Oil-Based Scent, 24-Hour Odor Protection, Cedar & Patchouli Blend, Gentle on
Sensitive Skin (Mountain Sage, 2.7 oz, 2-Pack)
\$15.95

Action: select_item()
Observation:
Subprocedure Done[I have selected the required item and it matches the
instruction.]
Current Page:
[Back to Search]
[< Prev]
```

---

```
scent [assorted scents][bright citrus][calming lavender][ginger fresh][simply
non-scents]
size [travel set (4-pack)][3 ounce (pack of 1)][3-ounce (2-pack)]
Bright Citrus Deodorant by Earth Mama | Natural and Safe for Sensitive Skin,
Pregnancy and Breastfeeding, Contains Organic Calendula 3-Ounce
Price: \$10.99
Rating: N.A.
[Description]
[Features]
[Reviews]
[Buy Now]

Action: select_item_attributes()
Observation:
Subprocedure Done[I have selected the additional attributes.]
Current Page:
You have clicked 3 ounce (pack of 1).

Action: purchase_item()

WebShop
Instruction:
i need some living room furniture, and price lower than 100.00 dollars

Action: think[I need to first search for available items, then select the item
that matches the instruction. If there are additional options for certain
attributes, I also need to select them before purchase. Otherwise I can
directly purchase the item.]
Observation: OK.

Action: search_item()
Observation:
Subprocedure Done[I have found a list of items.]
Current Page:
Observation:
[Back to Search]
Page 1 (Total results: 50)
[Next >]
[B09G71YGSV]
Patterned Square Pouf in Black and Gray Boho Decor Bean Bag Bedroom Furniture
Storage Ottoman Accent Chair Bean Bag Chair Papasan Chair Living Room Rugs
Floor Pillow Bedroom Chair Foot Stool
\$100.0
[B00UNYEOVE]
TOV Furniture The Gavin Collection Modern Tufted Linen Fabric Upholstered
Living Room Sofa, Beige, 85.25" x 34.5" x 27.75"
\$100.0
[B07194KS6G]
Hekman Furniture Round Coffee Table
\$100.0

Action: select_item()
Observation:
Subprocedure Done[I have selected the required item and it matches the
instruction.]
Current Page:
[Back to Search]
[< Prev]
TOV Furniture The Gavin Collection Modern Tufted Linen Fabric Upholstered
Living Room Sofa, Beige, 85.25" x 34.5" x 27.75"
Price: \$100.0
Rating: N.A.
```

```
[Description]
[Features]
[Reviews]
[Attributes]
[Buy Now]

Action: purchase_item()
```

Below is a template of the prompt for skill-conditioned policies. Then we provide an example with skill *select_item_attributes()*.

---

**WebShop: skill-conditioned policy prompt template**

```
Interact with a webshop to solve a task. Please consider the following tips to
solve new tasks:

‘‘‘
{policy improvement tips}
{primitives}
‘‘‘

Here are some examples.

‘‘‘
{examples}
‘‘‘
```

---

**WebShop: skill-conditioned policy prompt for** *select_item_attributes()*

```
Interact with a webshop to solve a task. Please consider the following tips to
solve new tasks:
1. You should strictly use the exact words of each clickable option without
making any changes.
2. Ensure to select all the necessary attributes as per the instruction.
3. Pay attention to the product details to ensure they match the instruction.
4. Be aware that color and size options may not always be straightforward and
may include additional information or codes.
5. Be careful with the case and spacing of the words when selecting item
attributes.
6. Make sure to select a product that has all the required attributes as per
the instruction.
7. Use quotation marks around specific phrases when searching for items to get
more accurate results.
8. Please only generate following actions in your response:
    click[Attribute]
    # Example click[38 m eu]
    click[Color]
    # Example click[brown]
    click[Style]
    # Example click[powerlite 1785w]
    click[Size]
    # Example click[8.5]
    click[Flavor]
    # Example click[plantain chips]
    click[Features]
    # Example click[Features]
    click[Digital Storage Capacity]
    # Example click[32 gb]
    click[Offer Type]
    # Example click[lockscreen ad-supported]
    click[Description]
    # Example click[Description]
```

---

```
Here are some examples.

Webshop
Instruction:
i would like a 3 ounce bottle of bright citrus deodorant for sensitive skin,
and price lower than 50.00 dollars
[Search]

Your historical interactions:
Action: search[3 ounce bright citrus deodorant sensitive skin]
Observation:
[Back to Search]
Page 1 (Total results: 50)
[Next >]
[B078GWRC1J]
Bright Citrus Deodorant by Earth Mama | Natural and Safe for Sensitive Skin,
Pregnancy and Breastfeeding, Contains Organic Calendula 3-Ounce
$10.99
[B078GTKVXY]
Ginger Fresh Deodorant by Earth Mama | Natural and Safe for Sensitive Skin,
Pregnancy and Breastfeeding, Contains Organic Calendula 3-Ounce
$10.99
[B08KBVJ4XN]
Barrel and Oak - Aluminum-Free Deodorant, Deodorant for Men, Essential
Oil-Based Scent, 24-Hour Odor Protection, Cedar & Patchouli Blend, Gentle on
Sensitive Skin (Mountain Sage, 2.7 oz, 2-Pack)
$15.95
Action: click[B078GWRC1J]
Observation:
[Back to Search]
[< Prev]
scent [assorted scents][bright citrus][calming lavender][ginger fresh][simply
non-scents]
size [travel set (4-pack)][3 ounce (pack of 1)][3-ounce (2-pack)]
Bright Citrus Deodorant by Earth Mama | Natural and Safe for Sensitive Skin,
Pregnancy and Breastfeeding, Contains Organic Calendula 3-Ounce
Price: $10.99
Rating: N.A.
[Description]
[Features]
[Reviews]
[Buy Now]

Your task is: select_attributes()

Action: think[For 3 ounce bottle of bright citrus deodorant for sensitive skin,
the item has options 'bright citrus' and '3 ounce (pack of 1)' and seems good
to buy.]
Observation: OK.

Action: click[bright citrus]
Observation: You have clicked bright citrus.

Action: click[3 ounce (pack of 1)]
Observation: You have clicked 3 ounce (pack of 1).

Action: Done[I have selected the additional attributes.]
```

E.2 Code-based Policy

**ALFWorld: Skill discovery**

```
You are a code generator.
```

You based on the history of agents interacting with the environment to generate a general code solution to solve the task.

```
**************************************************************************
You have access to the following primitive functions:
# example: go to table 1 => go('table 1')
def go(location: str):
    return 'go to ' + location

# example: open drawer 1 => open('drawer 1')
def open(object: str):
    return 'open ' + object

# example: The cabinet 5 is closed => isclose('The cabinet 5 is closed') return
True
def isclose(observation: str):
    if "is closed" in observation:
        return True
    else:
        return False

# example: input_str = "cabinet 3, coffeemachine 1,cabinet 5,", object_name
="coffeemachine", return "coffeemachine 1"
def extract_pattern(input_str, object_name):
    # Using RegEx pattern to find the object_name followed by a space and a
number
    pattern = r'{} \d+'.format(object_name)
    match = re.search(pattern, input_str)

    # If a match is found
    if match:
        return match.group()  # This returns the first match found
    else:
        return "No match found."

def interact_with_environment(action):
    env_last_obs, _, _, info = env.step([action])
    print(action)
    print(env_last_obs)
    return env_last_obs[0],info['won'][0]

def is_in(object,env_last_obs):
    # check whether an object is in the envriて observation
    if object in env_last_obs:
        return True
    return False

# example: use desklamp 1 => use('desklamp 1')
def use(desklamp: str):
    return 'use ' + desklamp

# example: close drawer 1 => close('drawer 1')
def close(object: str):
    return 'close ' + object

# example: put spraybottle 1 in/on toilet 1 => put('spraybottle 1', 'toilet 1')
def put(object: str, location: str):
    return 'put ' + object + ' in/on ' + location

# example: clean cloth 1 with sinkbasin 1  => clean('cloth 1','sinkbasin 1')
def clean(object_1: str, sinkbasin: str):
    return 'clean ' + object_1 + ' with ' + sinkbasin
```

```
# example: take keychain 2 from sofa 1 => take('keychain 2', 'sofa 1')
def take(object: str, location: str):
    return 'take ' + object + ' from ' + location

# example: heat tomato 2 with microwave 1
def heat(object: str, microwave: str):
    return 'heat ' + object + ' with ' + microwave

# example: cool pot 1 with fridge 1
def cool(object: str, fridge: str):
    return 'cool ' + object + ' with ' + fridge

**************************************************************************
Tips:
1. The basic interaction can be in the following format:
action = go(location)
env_observation = interact_with_environment(action)

2. You can extract specific object names by:
specific_object = extract_pattern(env_observation, object_to_find)

I will start showing you the interaction history of task find.
You can summarize new tips based on the new interaction and generate the
function based on the tips and primitives.
The function should be in the format of

def clean(object_to_heat):
    # You can assume the object is already with you.
    # Your implementation here
```

**WebShop: Skill discovery**

```
You are a code generator.
You based on the history of agent interacting with the environment to generate
a general code solution to solve the task.

**************************************************************************
You have access to the following primitive functions:

# Example select(B0054KM7IY)
def select(item_number):
    return f'click[{item_number}]'

# Example buy()
def buy():
    return f'click[Buy Now]'

# Example:
# click(brown)
# click(Flavor)
# click(rectangular)
# click(Description)
# click(32 gb)
def click(feature):
    return f'click[{feature}]'

def interact_with_environment(action):
    global env_id
    env_obs,reward,done = env.step(env_id,action)
    print(action)
    print(env_obs)
```

```
    return env_obs, reward

def search(instruction):
    #Usage: search item based on the instruction
    #Example #1:
    #Instruction:  im looking for a light pink long handle back loofah shower
brush, and price lower than 40.00 dollars
    #Return: search[light pink long handle back loofah shower brush]

def extract_requirement(instruction):
    #Usage: based on instruction, extract requirements
    #Instruction:  im looking for a light pink long handle back loofah shower
brush, and price lower than 40.00 dollars
    #Return: a list of requirement ['easy apply','pine tar scented','price
lower than 50.00 dollars']

def requirement_match_check(item_discription,requirement):
    #Usage: Check whether an item discription (not item attributes) satisfy
certain requirement
    #item_discription = 'JulaJuyo Back Scrubber for Shower, Long Handle Body
Bath Brush Showering Loofah Sponge on a Stick for Men Women, Nylon Mesh
Exfoliating Bathing Lufa Scrub Shower Cleaning Luffa Brush (1 Pack-White)
\$7.99 '
    #requirement = 'light pink'
    #return  Ture/False

def parse_items_observation_into_items_dict(items_observation):
    #Usage: parse a string of item into a dictionary.
    #items_observation="[B09GLCXKP6] ApexDesk MK Series Children's Height
Adjustable Chair with Study Desk w/Integrated Shelf & Drawer (Pink Desk & MK
Chair Bundle) \$419.99
    #[B0184JRW1I] ApexDesk Little Soleil DX 43" Children's Height Adjustable
Study Desk w/ Integrated Shelf & Drawer (Desk+Chair Bundle { Blue) \$359.99 "
    # return {'B09GLCXKP6':'ApexDesk MK...','B0184JRW1I': 'ApexDesk Little...'}

def select_most_relevant_attibutes(item_description,requirements):
    #Usage: return a list of most relevant attributes you can directly click
based on item_description
    #item_description: string
    #requirements: [requirements#1,requirements#2,requirements#3...]
    return [attribute#1,attributes#2..]

**************************************************************************
Tips:
1. The basic interaction can be in the following format:
env_observation,reward = interact_with_environment(action)
2. You should focus on to generate the code for the interaction after "Your
task is:select_item()", i.e why the Action is executed (after "Your task
is:select_item()")?
3. You need to based on the interaction history before "Your task
is:select_item()" to infer why the Action is executed
4. It is a web shopping interface, you are browsing web pages.
5. You are allowed to use numpy library: import numpy as np
6. You code should be able to generate action based on the observation and
interact with env with interact_with_environment(action)

I will start show you the interaction history of task find.
You can summarize new tips based on the new interaction and generate the
function based on the tips and primitives.
The function should be in the format of
```

```
def select_item(items,instruction):
    # items: dict {{'item number': discription}}
    # instruction: str

    # Hints:
    1. You may maintain a dictionary where key is the item number and the
values containing number of satified requirements
    2. Never select or click [<Prev] and [Next>]. Do not evenwrite [<Prev] and
[Next>] in your code.

    # return the best match item number among all items. no need to match all
requirements in the instruction
```

We encapsulate the LLM within the specified functions, exposing only the function signature and usage in the aforementioned prompt to the LLM.

**Webshop: Skill discovery**

```
def extract_requirement(instruction):
    user_content = f"""
    Given an instruction, you need decompose it and find the requirments:
    For example:

    Example #1:
    Instruction:
    i'm looking for a light pink long handle back loofah shower brush, and
price lower than 40.00 dollars

    You return:
    requirement_1 = 'light pink'
    requirement_2 = 'long handle back loofah'
    requirement_3 =  'price lower than 40.00 dollars'

    Example #2:
    Instruction:
    i need easy apply pine tar scented mustache wax stick for men, and price
lower than 50.00 dollars

    You return:
    {{'requirement_1' : 'easy apply'
    'requirement_2': 'pine tar scented'
    'requirement_3':'price lower than 50.00 dollars'}}

    Now given an new instruction:
    {instruction}
    You return:
    """
    res, _ = query_openai_api(user_content=user_content)
    return res

def requirement_match_check(item,requirement):
    requirement_embedding = get_embedding(requirement)
    item_embedding = get_embedding(item)
    score = cosine_similarity(requirement_embedding,item_embedding)
    print("similarity score",score)
    if score>0.8:
        return True
    else:
        return False

def select_most_relevant_attibutes(item_description,requirement):
```

```
    user_content = """
    For example:

    Example #1:
    Given item_description:
    scent [citrus & spice][essential 7][extra firm hold][lime & sage][mountain
fresh][original blend][pine tar][unscented]
    Mountaineer Brand Stache Stick - All-Natural Convenient 1.5 Oz Mustache Wax
Stick for Men (Extra Firm Hold)
    Price: \$16.99
    Rating: N.A.
    [Description]
    [Features]
    [Reviews]
    [Attributes]
    [Buy Now]

    And requirements:
    requirement_1 = 'light pink'
    requirement_2 = 'long handle back loofah'
    requirement_3 =  'price lower than 40.00 dollars'

    You return:
    [[pine tar]]

    Example #2:
    Given item_description:
    scent [2 pack-beige][2 pack-green][2 pack-light pink][2
pcs=green+beige][2 pcs=green+pink][3 pack-beige][3 pack-light pink][3
pcs=colors][beige][green][light pink]
    Prozklves Back Scrubber for Shower, Long Handle Back Loofah Shower Brush,
Soft Nylon Mesh Back Cleaner Washer, Bath Brush for Women Men, Exfoliating Body
Scrubber for Elderly (Light Pink)
    Price: \$7.99
    Rating: N.A.
    [Description]
    [Features]
    [Reviews]
    [Attributes]
    [Buy Now]

    And requirements:
    requirement_1 = 'light pink'
    requirement_2 = 'long handle back loofah'
    requirement_3 =  'price lower than 40.00 dollars'

    You return:
    [[light pink]]
    """
```

### E.3 Human-Generated Policy Improvement Tips and Primitives for Each Skill

#### E.3.1 ALFWorld

**ALFWorld: Policy improvement tips and primitives for skill clean(object)**

```
1. Do not try to turn on the sinkbasin; it does not work.
2. You can directly clean an object with a sinkbasin by taking the
corresponding action.
3. Avoid going to irrelevant places when you are not sure about the next action;
it wastes time and does not help in accomplishing the task.
```

```
4. Focus on the task at hand and avoid unnecessary actions.
5. Check the contents of a location before moving to another one.
6. Attempt to perform the task as soon as you have the necessary item and are
in the correct location.
7. You can think about your plan in the first step, and after that you can only
select from the following actions in your response (please strictly follow the
syntax):
    go to [receptacle] [id]
    # Example:
        go to countertop 1
    clean [object] [id] with [receptacle] [id]
    # Example:
        clean apple 1 with sinkbasin 1
```

**ALFWorld: Policy improvement tips and primitives for skill cool(object)**

```
1. Please complete the task with as few steps as possible.
2. You can think about your plan in the first step, and after that you can only
select from the following actions in your response (please strictly follow the
syntax):
    go to [receptacle] [id]
    # Example:
        go to countertop 1
    cool [object] [id] with [receptacle] [id]
    # Example:
        cool egg 1 with fridge 1
3. You can take action cool [object] [id] with [receptacle] [id] even when the
receptacle is closed.
```

**ALFWorld: Policy improvement tips and primitives for skill find(object)**

```
1. Try to execute a different action when you stuck in a loop of taking
identical actions.
2. Try to search all other locations if the object could not be found in
initial likely locations.
3. Remember the location of each object you have seen in order to quickly find
it later.
4. Remember the number of objects you have seen in each location, so you can
directly go back the location to find another one.
5. Do not mismatch target obejct with similar object, for example, a mug is
different with a cup.
6. Please complete the task with as few steps as possible.
7. You can think about your plan in the first step, and after that you can only
select from the following actions in your response (please strictly follow the
syntax):
    go to [receptacle] [id]
    # Example:
        go to countertop 1
    open [receptacle] [id]
    # Example:
        open drawer 2
```

**ALFWorld: Policy improvement tips and primitives for skill heat(object)**

```
1. You need to first go to a microwave to heat an object.
2. You can heat any object with microwave.
3. Do not open the microwave, directly take the heat operation.
5. Only select from the following actions in your response (please strictly
follow the syntax):
go to [receptacle] [id]
# Example:
    go to countertop 1
heat [object] [id] with [receptacle] [id]
```

```
# Example:
    heat egg 1 with microwave 1
```

**ALFWorld: Policy improvement tips and primitives for skill put(object, receptacle)**

```
1. Try to execute a different action when you stuck in a loop of taking
identical actions.
2. You can directly put an object in/on an occupied receptacle.
3. Please complete the task with as few steps as possible.
4. You can think about your plan in the first step, and after that you can only
select from the following actions in your response (please strictly follow the
syntax):
    go to [receptacle] [id]
    # Example:
        go to countertop 1
    open [receptacle] [id]
    # Example:
        open drawer 2
    put [object] [id] in/on [receptacle] [id]
    # Example:
        put lettuce 1 in/on countertop 1
```

**ALFWorld: Policy improvement tips and primitives for skill take(object)**

```
1. Try to execute a different action when you stuck in a loop of taking
identical actions.
2. You can think about your plan in the first step, and after that you can only
select from the following actions in your response (please strictly follow the
syntax):
    take [object] [id] from [receptacle] [id]
    # Example:
        take lettuce 1 from countertop 1
```

**ALFWorld: Policy improvement tips and primitives for skill use(object)**

```
1. Try to execute a different action when you stuck in a loop of taking
identical actions.
2. Please complete the task with as few steps as possible.
3. You can think about your plan in the first step, and after that you can only
select from the following actions in your response (please strictly follow the
syntax):
    use [receptacle] [id]
    # Example:
        use desklamp 1
```

### E.3.2 WebShop

**WebShop: Policy improvement tips and primitives for skill search_item()**

```
1. Please pay attention to instruction description to find the product that
exactly matches with the requirement in the instruction;
2. You should pay attention to the current observation to check clickable
buttons;
3. When the current selected product does not match the requirement in the
instruction well, you should navigate to the results page to check other likely
product candidates;
4. When there is no product that matches the requirement in the instruction
well, you should search again with a new query;
5. You should click[Back to Search] before re-searching with a new query;
6. Please only generate following actions in your response:
    search[Query]
    # Example search[3 ounce bright citrus deodorant sensitive skin]
```

```
    click[Back to Search]
    # Example click[Back to Search]
```

---

**WebShop: Policy improvement tips and primitives for skill select_item()**

```
1. Please pay attention to instruction description to find the product that
exactly matches with the requirement in the instruction;
2. You can click[Description] or click[Features] to find the detailed
description of a product;
3. You should pay attention to the current observation to check clickable
buttons;
4. When the current selected product does not match the requirement in the
instruction well, you should navigate to the results page to check other likely
product candidates;
5. Please only generate following actions in your response:
    click[Item ID]
    # Example click[B0054KM7IY]
    click[< Prev]
    # Example click[< Prev]
```

---

**WebShop: Policy improvement tips and primitives for skill select_item_attributes()**

```
1. Please pay attention to instruction description to find the product that
exactly matches with the requirement in the instruction;
2. You can click[Description] to find the detailed description of a product;
3. Remember to select all required attributes using click[Attributes] before
click[Buy Now];
4. You should click[Description] or click[Features] to double-check attributes
when there is no clickable attributes;
5. You should pay attention to the current observation to check clickable
buttons;
6. Please only generate following actions in your response:
    click[Attributes]
    # Example 1: click[16.9 fl oz (pack of 1)]
    # Example 2: click[blue]
    click[< Prev]
    # Example: click[< Prev]
    click[Description]
    # Example: click[Description]
    click[Features]
    # Example: click[Features]
```

---

**WebShop: Policy improvement tips and primitives for skill purchase_item()**

```
1. Please only generate following actions in your response:
    click[Buy Now]
    # Example click[Buy Now]
```

## E.4    LLM-Distilled Policy Improvement Tips and Primitives for Each Skill

### E.4.1    ALFWorld

---

**ALFWorld: Policy improvement tips and primitives for skill clean(object)**

```
1. Do not try to turn on the sinkbasin; it does not work.
2. You can directly clean an object with a sinkbasin by taking the
corresponding action.
3. Avoid going to irrelevant places when you are not sure about the next action;
it wastes time and does not help in accomplishing the task.
4. Focus on the task at hand and avoid unnecessary actions.
5. Check the contents of a location before moving to another one.
```

```
6. Attempt to perform the task as soon as you have the necessary item and are
in the correct location.
7. You can think about your plan in the first step, and after that you can only
select from the following actions in your response (please strictly follow the
syntax):
    go to [receptacle] [id]
    # Example:
        go to countertop 1
    clean [object] [id] with [receptacle] [id]
    # Example:
        clean apple 1 with sinkbasin 1
```

**ALFWorld: Policy improvement tips and primitives for skill cool(object)**

```
1. Always remember to locate and pick up the object first before trying to
perform an action on it.
2. An object can be cooled using a fridge.
3. You can directly cool an object with a fridge by taking the corresponding
action.
4. Avoid getting stuck in a loop of actions that do not contribute to the
completion of the task.
5. Explore the environment thoroughly to locate the object needed to complete
the task.
6. If an object is not in its usual place, check other possible locations.
7. If you can't find an object in one location, it might be in another. In this
case, the lettuce was on the countertop, not the fridge.
8. Avoid attempting actions that are not possible in the given environment.
9. You can think about your plan in the first step, and after that you can only
select from the following actions in your response (please strictly follow the
syntax):
    go to [receptacle] [id]
    # Example:
        go to countertop 1
    cool [object] [id] with [receptacle] [id]
    # Example:
        cool egg 1 with fridge 1
```

**ALFWorld: Policy improvement tips and primitives for skill find(object)**

```
1. Thoroughly check each location before moving on to the next.
2. Do not pick up unnecessary items.
3. Do not waste time checking locations where the item is unlikely to be found.
4. Check all possible locations when looking for an item.
5. Don't assume an item won't be in a certain location based on where similar
items were found.
6. If you can't find an item, try to revisit the locations you have already
checked. The item might be hidden or covered by other items.
7. Do not exclude any locations when searching for an item, even if they seem
unlikely.
8. Expand your search area when you can't find an item in the expected
locations.
9. Remember that items can be found in unexpected places.
10. Avoid trying to take an item from a location where it has not been found.
11. Revisit locations where similar items were found as they might contain the
item you are looking for.
12. Always check all possible locations, even if they seem unlikely to contain
the item.
13. Remember to check the dining table when looking for items like a remote
control.
14. You can think about your plan in the first step, and after that you can
only select from the following actions in your response (please strictly follow
the syntax):
    go to [receptacle] [id]
```

```
    # Example:
        go to countertop 1
open [receptacle] [id]
    # Example:
        open drawer 2
```

**ALFWorld: Policy improvement tips and primitives for skill heat(object)**

```
1. Try to execute a different action when you get stuck in a loop of taking
identical actions.
2. Ensure you have the correct object in your possession before attempting to
perform actions on it.
3. Check all possible locations if you can't find an object in the expected
places.
4. You can directly heat an object with a microwave by taking the corresponding
action.
5. If an action does not work, try a different approach or command.
6. Focus on relevant locations to complete the task efficiently.
7. A cup is not the same as a mug. Make sure to find and use the correct object
for the task.
8. You can think about your plan in the first step, and after that you can only
select from the following actions in your response (please strictly follow the
syntax):
    go to [receptacle] [id]
    # Example:
        go to countertop 1
    heat [object] [id] with [receptacle] [id]
    # Example:
        heat egg 1 with microwave 1
```

**ALFWorld: Policy improvement tips and primitives for skill put(object, receptacle)**

```
1. When looking for an object, make sure to check all possible locations.
2. If an object is not in the expected location, it may be somewhere else in
the room.
3. Avoid getting stuck in a loop of unsuccessful actions. If an action doesn't
work, try a different approach.
4. Not all surfaces may be suitable for placing objects. If one doesn't work,
try another.
5. Focus on the task at hand and avoid unnecessary actions, such as searching
through irrelevant locations or performing unnecessary actions.
6. Use the correct preposition when placing an object on a surface. The correct
action is to put the object "in/on" the surface, not just "on" the surface.
7. Do not attempt to modify the object if it is not necessary for the task.
8. When placing an object on a surface, make sure the surface is not already
full. If it is, try a different surface.
9. After picking up an object, perform the necessary action immediately instead
of moving around unnecessarily.
10. If you already have the object needed for the task, do not try to find more
of the same object.
11. Place the object in the first available and suitable location instead of
unnecessarily checking other locations.
12. Use the correct command to put an object in a cabinet. The correct command
is "put [object] in/on [surface]".
13. After cleaning an object, if the task is to put it in a drawer, do it
immediately instead of checking other drawers.
14. If you put an object in a device (like a microwave), remember to take it
out before trying to put it in another location.
15. Do not attempt to put an object on top of the fridge. The correct action is
to put the object "in/on" the fridge.
16. Do not attempt to put an object on top of other objects on a surface. The
correct action is to put the object directly "in/on" the surface.
```

```
17. You can think about your plan in the first step, and after that you can
only select from the following actions in your response (please strictly follow
the syntax):
    go to [receptacle] [id]
    # Example:
        go to countertop 1
    open [receptacle] [id]
    # Example:
        open drawer 2
    put [object] [id] in/on [receptacle] [id]
    # Example:
        put lettuce 1 in/on countertop 1
```

**ALFWorld: Policy improvement tips and primitives for skill take(object)**

```
1. Ensure the object you want to take is actually in the location you are
trying to take it from.
2. Do not get stuck in a loop of trying to take an object from a location where
it is not present.
3. Avoid unnecessary actions that do not contribute to the completion of the
task.
4. Be aware of your current location before attempting to take an object.
5. Explore different locations if you cannot find the object in the initial
locations you check.
6. If you can't find an object, try to look for it in other possible locations.
7. Don't waste time checking empty or irrelevant locations. Focus on the
locations where the object is likely to be found.
8. Pay attention to the task description and take the correct object.
9. Don't confuse similar objects. Make sure to take the correct object as per
the task description.
10. Check all possible locations before attempting to take an object.
11. Avoid unnecessary actions that are not related to the task, such as trying
to put an object on various shelves when the task is to take an object.
12. Laptops can also be found on sofas, not just on tables or in drawers.
13. You can think about your plan in the first step, and after that you can
only select from the following actions in your response (please strictly follow
the syntax):
    take [object] [id] from [receptacle] [id]
    # Example:
        take lettuce 1 from countertop 1
```

**ALFWorld: Policy improvement tips and primitives for skill use(object)**

```
1. Stay focused on the task and avoid unnecessary actions that are not related
to the task.
2. Some objects can be used directly without needing to be moved or interacted
with other objects.
3. If an action doesn't work, it's likely not necessary for the task. Try a
different approach.
4. Avoid getting stuck in a loop of unproductive actions. If an action doesn't
produce a result, it's unlikely to do so with repeated attempts.
5. Don't assume that a task requires complex actions or interactions between
objects. Sometimes, the solution is straightforward.
6. You can think about your plan in the first step, and after that you can only
select from the following actions in your response (please strictly follow the
syntax):
    use [receptacle] [id]
    # Example:
        use desklamp 1
```

E.4.2   WebShop

---

**WebShop: Policy improvement tips and primitives for skill purchase_item()**

```
1. You should strictly use the exact words of each clickable option without
making any changes.
2. Always remember to select the attributes of an item before purchasing it.
3. Ensure to select the correct color, size, and style options as per the task
instruction before purchasing an item.
4. Do not rush to purchase an item without selecting its attributes.
5. Carefully read and understand the task instruction before making a
selection.
6. Always ensure that the options you want to select are available on the page
before attempting to click on them.
7. Always select all the necessary attributes of an item as per the task
instruction before purchasing it.
8. Always consider the price limit in the instruction when selecting a product.
9. Use simplified and more general terms when searching for items to get more
accurate results.
10. Please only generate following actions in your response:
    click[Buy Now]
    # Example click[Buy Now]
```

---

**WebShop: Policy improvement tips and primitives for skill search_item()**

```
1. Use specific search terms that include all the essential details of the item
you are looking for.
2. Use the exact terminology when performing a search to get the most accurate
results.
3. Always include all the necessary details in the search query to get the most
accurate results.
4. Use quotation marks around specific search terms to get the most accurate
search results.
5. Avoid being overly specific in your search terms as it may limit the search
results.
6. Please only generate following actions in your response:
    search[Query]
    # Example search[3 ounce bright citrus deodorant sensitive skin]
    click[Back to Search]
    # Example click[Back to Search]
```

---

**WebShop: Policy improvement tips and primitives for skill select_item_attributes()**

```
1. You should strictly use the exact words of each clickable option without
making any changes.
2. Ensure to select all the necessary attributes as per the instruction.
3. Pay attention to the product details to ensure they match the instruction.
4. Be aware that color and size options may not always be straightforward and
may include additional information or codes.
5. Be careful with the case and spacing of the words when selecting item
attributes.
6. Make sure to select a product that has all the required attributes as per
the instruction.
7. Use quotation marks around specific phrases when searching for items to get
more accurate results.
8. Please only generate following actions in your response:
    click[Attribute]
    # Example click[38 m eu]
    click[Color]
    # Example click[brown]
    click[Style]
    # Example click[powerlite 1785w]
    click[Size]
    # Example click[8.5]
    click[Flavor]
```

```
# Example click[plantain chips]
click[Features]
# Example click[Features]
click[Digital Storage Capacity]
# Example click[32 gb]
click[Offer Type]
# Example click[lockscreen ad-supported]
click[Description]
# Example click[Description]
```

---

**WebShop: Policy improvement tips and primitives for skill select_item()**

```
1. You should strictly use the exact words of each clickable option without
making any changes.
2. When searching for items with specific characteristics, use quotation marks
around the specific characteristic to narrow down the search results.
3. Review multiple items before making a selection to ensure the chosen item
best fits the given criteria.
4. After finding the desired item, make sure to complete the task by selecting
the item.
5. Avoid over-specifying the search terms to get more relevant results.
6. Always ensure to complete the task as per the instructions given.
7. Pay attention to the item details in the search results to ensure they match
the given criteria.
8. Ensure the selected item strictly meets the given criteria.
9. Include all the criteria in the search query to narrow down the search
results and find the most suitable item more efficiently.
10. Confirm the specific characteristics of the item by checking the item
details.
11. Always consider the price limit when selecting an item.
12. After clicking on an item, make sure to select it to complete the task.
13. Please only generate following actions in your response:
    click[Item ID]
    # Example click[B0054KM7IY]
    click[< Prev]
    # Example click[< Prev]
```

---

## E.5 Ablation Study and Discussion on Policy Improvement Tips

### E.5.1 Tip Distillation Only (TD-Only) Prompts

---

**ALFWorld: Contrastive tip distillation prompt**

```
You will be given both failure and success histories in which you were placed
in an environment and given a task to complete. By comparing the two histories,
please find the mistakes in the failure trial, and summarize general tips
to fix the mistakes. The tips can also be about the rules in this particular
environment. You will need these tips later to solve a similar task. When there
is already a list of tips to start with, please keep them in your response
or improve them, and you can also append new tips according to the given
new trials. You should not add duplicate tips. Give your response after "New
summarization: "

Here is one example:

Task 1:
Failure trial:
You are in the middle of a room. Looking quickly around you, you see a cabinet
10, a cabinet 9, a cabinet 8, a cabinet 7, a cabinet 6, a cabinet 5, a cabinet
4, a cabinet 3, a cabinet 2, a cabinet 1, a coffeemachine 1, a countertop
2, a countertop 1, a diningtable 2, a diningtable 1, a drawer 3, a drawer
2, a drawer 1, a fridge 1, a garbagecan 1, a microwave 1, a sinkbasin 1, a
```

```
stoveburner 4, a stoveburner 3, a stoveburner 2, a stoveburner 1, and a toaster
1.
Your task is to: put a hot mug on a countertop.
> go to countertop 1
On the countertop 1, you see a butterknife 1, a cup 1, a fork 2, and a mug 1.
> take mug 1 from countertop 1
You pick up the mug 1 from the countertop 1.
> go to stove burner 1
On the stoveburner 1, you see a pan 2.
> heat mug 1 with stoveburner 1
Nothing happens.
> heat mug 1 with stoveburner 1
Nothing happens.

Success trial:
You are in the middle of a room. Looking quickly around you, you see a cabinet
10, a cabinet 9, a cabinet 8, a cabinet 7, a cabinet 6, a cabinet 5, a cabinet
4, a cabinet 3, a cabinet 2, a cabinet 1, a coffeemachine 1, a countertop
2, a countertop 1, a diningtable 2, a diningtable 1, a drawer 3, a drawer
2, a drawer 1, a fridge 1, a garbagecan 1, a microwave 1, a sinkbasin 1, a
stoveburner 4, a stoveburner 3, a stoveburner 2, a stoveburner 1, and a toaster
1.
Your task is to: put a hot mug on a countertop.
> go to countertop 1
On the countertop 1, you see a butterknife 1, a cup 1, a fork 2, and a mug 1.
> take mug 1 from countertop 1
You pick up the mug 1 from the countertop 1.
> go to microwave 1
You arrive at loc 22. The microwave 1 is closed.
> heat mug 1 with microwave 1
You heat the mug 1 using the microwave 1.
> go to countertop 1
On the countertop 1, you see a butterknife 1, a cup 1, a fork 2.
> put mug 1 in/on countertop 1
You put the mug 1 in/on the countertop 1

New summarization: In task 1, I made a mistake in heating a mug with a
stoveburner, and I got stuck in a loop where I continually did this. As shown
in the success trial, the correct way is to heat a mug with a microwave. Thus,
I should try different actions when getting stuck in a loop; I should use a
microwave to heat a mug, and I can directly heat an object with a microwave
without opening the microwave and putting the object in it.

Tips:
- Try to execute a different action when you get stuck in a loop of taking
identical actions.
- Mug should be heated by a microwave rather than a stoveburner.
- You can directly heat an object with a microwave by taking the corresponding
action.
```

---

**ALFWorld: Non-contrastive tip distillation prompt**

```
You will be given success histories in which you were placed in an environment
and given a task to complete. Please summarize general tips according to the
success histories. The tips can be the rules in this particular environment.
You will need these tips later to solve a similar task. When there is already a
list of tips to start with, please keep them in your response or improve them,
and you can also append new tips according to the given new trials. You do not
have to add new tips if there is no one. You should not add duplicate tips.
Give your response after "New summarization: "
```

```
Here is one example:

Task 1:
Success trial:
You are in the middle of a room. Looking quickly around you, you see a cabinet
10, a cabinet 9, a cabinet 8, a cabinet 7, a cabinet 6, a cabinet 5, a cabinet
4, a cabinet 3, a cabinet 2, a cabinet 1, a coffeemachine 1, a countertop
2, a countertop 1, a diningtable 2, a diningtable 1, a drawer 3, a drawer
2, a drawer 1, a fridge 1, a garbagecan 1, a microwave 1, a sinkbasin 1, a
stoveburner 4, a stoveburner 3, a stoveburner 2, a stoveburner 1, and a toaster
1.
Your task is to: put a hot mug on a countertop.
> go to countertop 1
On the countertop 1, you see a butterknife 1, a cup 1, a fork 2, and a mug 1.
> take mug 1 from countertop 1
You pick up the mug 1 from the countertop 1.
> go to microwave 1
You arrive at loc 22. The microwave 1 is closed.
> heat mug 1 with microwave 1
You heat the mug 1 using the microwave 1.
> go to countertop 1
On the countertop 1, you see a butterknife 1, a cup 1, a fork 2.
> put mug 1 in/on countertop 1
You put the mug 1 in/on the countertop 1

New summarization: In task 1, as shown in the success trial, the correct way
is to heat a mug with a microwave. Thus, I should use a microwave to heat a
mug, and I can directly heat an object with a microwave without opening the
microwave and putting the object in it.

Tips:
- Mug should be heated by a microwave rather than a stoveburner.
- You can directly heat an object with a microwave by taking the corresponding
action.
```

**WebShop: Contrastive tip distillation prompt**

```
You will be given both failure and success histories in which you were placed
in an environment and given a task to complete. By comparing the two histories,
please find the mistakes in the failure trial, and summarize general tips
to fix the mistakes. The tips can also be about the rules in this particular
environment. You will need these tips later to solve a similar task. When there
is already a list of tips to start with, please keep them in your response
or improve them, and you can also append new tips according to the given new
trials. You should not add duplicate tips. You do not have to add new tips if
there is no one. Give your response after "New summarization: "

Here is one example:
Task1:
Failure Trial:
Webshop
Instruction:
i would like a 3 ounce bottle of bright citrus deodorant for sensitive skin,
and price lower than 50.00 dollars
[Search]

Action: search[3 ounce bright citrus deodorant sensitive skin]
Observation:
[Back to Search]
Page 1 (Total results: 50)
[Next >]
[B078GWRC1J]
```

```
Bright Citrus Deodorant by Earth Mama | Natural and Safe for Sensitive Skin,
Pregnancy and Breastfeeding, Contains Organic Calendula 3-Ounce
\$10.99
[B078GTKVXY]
Ginger Fresh Deodorant by Earth Mama | Natural and Safe for Sensitive Skin,
Pregnancy and Breastfeeding, Contains Organic Calendula 3-Ounce
\$10.99
[B08KBVJ4XN]
Barrel and Oak - Aluminum-Free Deodorant, Deodorant for Men, Essential
Oil-Based Scent, 24-Hour Odor Protection, Cedar & Patchouli Blend, Gentle on
Sensitive Skin (Mountain Sage, 2.7 oz, 2-Pack)
\$15.95

Action: click[B078GWRC1J]
Observation:
[Back to Search]
[< Prev]
scent [assorted scents][bright citrus][calming lavender][ginger fresh][simply
non-scents]
size [travel set (4-pack)][3 ounce (pack of 1)][3-ounce (2-pack)]
Bright Citrus Deodorant by Earth Mama | Natural and Safe for Sensitive Skin,
Pregnancy and Breastfeeding, Contains Organic Calendula 3-Ounce
Price: \$10.99
Rating: N.A.
[Description]
[Features]
[Reviews]
[Buy Now]

Action: click[BRIGHT CITRUS]
Observation: Invalid action!

Action: click[BRIGHT CITRUS]
Observation: Invalid action!

Action: click[BRIGHT CITRUS]
Observation: Invalid action!

Success Trial:
Webshop
Instruction:
i would like a 3 ounce bottle of bright citrus deodorant for sensitive skin,
and price lower than 50.00 dollars
[Search]

Action: search[3 ounce bright citrus deodorant sensitive skin]
Observation:
[Back to Search]
Page 1 (Total results: 50)
[Next >]
[B078GWRC1J]
Bright Citrus Deodorant by Earth Mama | Natural and Safe for Sensitive Skin,
Pregnancy and Breastfeeding, Contains Organic Calendula 3-Ounce
\$10.99
[B078GTKVXY]
Ginger Fresh Deodorant by Earth Mama | Natural and Safe for Sensitive Skin,
Pregnancy and Breastfeeding, Contains Organic Calendula 3-Ounce
\$10.99
[B08KBVJ4XN]
Barrel and Oak - Aluminum-Free Deodorant, Deodorant for Men, Essential
Oil-Based Scent, 24-Hour Odor Protection, Cedar & Patchouli Blend, Gentle on
Sensitive Skin (Mountain Sage, 2.7 oz, 2-Pack)
```

```
\$15.95

Action: click[B078GWRC1J]
Observation:
[Back to Search]
[< Prev]
scent [assorted scents][bright citrus][calming lavender][ginger fresh][simply
non-scents]
size [travel set (4-pack)][3 ounce (pack of 1)][3-ounce (2-pack)]
Bright Citrus Deodorant by Earth Mama | Natural and Safe for Sensitive Skin,
Pregnancy and Breastfeeding, Contains Organic Calendula 3-Ounce
Price: \$10.99
Rating: N.A.
[Description]
[Features]
[Reviews]
[Buy Now]

Action: click[bright citrus]
Observation: You have clicked bright citrus.

Action: click[3 ounce (pack of 1)]
Observation: You have clicked 3 ounce (pack of 1).

Action: click[Buy Now]

New summarization: In task 1, I get stuck in a loop where I continued taking an
invalid action click[BRIGHT CITRUS]. Thus, I should try different actions when
getting stuck in a loop; I also made a mistake on capitalizing the name of a
scent option [bright citrus], so that click[BRIGHT CITRUS] become invalid. As
shown in the success trial, the correct action should be click[bright citrus].
Thus, I should strictly use the exact words of each option without making any
changes.

Tips:
- Try to execute a different action when you get stuck in a loop of taking
identical actions.
- You should strictly use the exact words of each clickable option without
making any changes.
```

---

**WebShop: Non-contrastive tip distillation prompt**

```
You will be given success histories in which you were placed in an environment
and given a task to complete. Please summarize general tips according to the
success histories. The tips can also be about the rules in this particular
environment. You will need these tips later to solve a similar task. When there
is already a list of tips to start with, please keep them in your response
or improve them, and you can also append new tips according to the given new
trials. You should not add duplicate tips. You do not have to add new tips if
there is no one. Give your response after "New summarization: "

Here is one example:
Task1:
Success Trial:
Webshop
Instruction:
i would like a 3 ounce bottle of bright citrus deodorant for sensitive skin,
and price lower than 50.00 dollars
[Search]

Action: search[3 ounce bright citrus deodorant sensitive skin]
```

```
Observation:
[Back to Search]
Page 1 (Total results: 50)
[Next >]
[B078GWRC1J]
Bright Citrus Deodorant by Earth Mama | Natural and Safe for Sensitive Skin,
Pregnancy and Breastfeeding, Contains Organic Calendula 3-Ounce
\$10.99
[B078GTKVXY]
Ginger Fresh Deodorant by Earth Mama | Natural and Safe for Sensitive Skin,
Pregnancy and Breastfeeding, Contains Organic Calendula 3-Ounce
\$10.99
[B08KBVJ4XN]
Barrel and Oak - Aluminum-Free Deodorant, Deodorant for Men, Essential
Oil-Based Scent, 24-Hour Odor Protection, Cedar & Patchouli Blend, Gentle on
Sensitive Skin (Mountain Sage, 2.7 oz, 2-Pack)
\$15.95

Action: click[B078GWRC1J]
Observation:
[Back to Search]
[< Prev]
scent [assorted scents][bright citrus][calming lavender][ginger fresh][simply
non-scents]
size [travel set (4-pack)][3 ounce (pack of 1)][3-ounce (2-pack)]
Bright Citrus Deodorant by Earth Mama | Natural and Safe for Sensitive Skin,
Pregnancy and Breastfeeding, Contains Organic Calendula 3-Ounce
Price: \$10.99
Rating: N.A.
[Description]
[Features]
[Reviews]
[Buy Now]

Action: click[bright citrus]
Observation: You have clicked bright citrus.

Action: click[3 ounce (pack of 1)]
Observation: You have clicked 3 ounce (pack of 1).

Action: click[Buy Now]

New summarization: In task 1, as shown in the success trial, I searched by
several keywords, selected a relevant item and clicked required attributes
before placing the order.

Tips:
- You should searched with keywords, select the relevant item to check and click
attributes according to the instruction.
```

### E.5.2   Results in ALFWorld

```
ALFWorld: Distilled policy improvement tips with contrastive prompt

Please consider the following tips to solve new tasks:

1. Try to execute a different action when you get stuck in a loop of taking
identical actions.
2. When stuck in a loop, try a different action or look for the item in a
different location.
3. Mug should be heated by a microwave rather than a stoveburner.
```

4. You can directly heat an object with a microwave by taking the corresponding action.
5. Remember to heat items in the microwave when required.
6. Use the correct command to place items on the dining table.
7. Use the correct commands to interact with items.
8. Search in all possible locations for the required item.
9. Check all possible locations for the required item, not just the most obvious ones.
10. Items can be found in unexpected places like the garbage can or microwave.
11. Remember to cool items in the fridge when required.
12. Remember to clean items when required before placing them in their final location.
13. You can clean items directly with the sinkbasin without needing to turn it on.
14. You don't need to move items to use them together.
15. Sometimes the simplest action is the correct one.
16. Perform tasks in the correct order.

---

**ALFWorld: Distilled policy improvement tips with non-contrastive prompt**

Please consider the following tips to solve new tasks:

1. Mug should be heated by a microwave rather than a stoveburner. You can directly heat an object with a microwave by taking the corresponding action. Bread can also be heated using a microwave.
2. After heating an object, it can be placed in the fridge for cooling. You can directly cool an object with a fridge by taking the corresponding action. After cooling a cup in the fridge, it can be placed back in the microwave.
3. Cellphones can be found on sidetables and beds. After finding a cellphone, place it in the dresser.
4. Toiletpaper can be found in drawers, not necessarily on toiletpaperhangers.
5. Cups can be found in microwaves, not necessarily on countertops or in cabinets.
6. Kettles can be found on stoveburners. After finding the kettles, place them in the cabinet.
7. Keychains can be found on shelves. After finding the keychains, place them in the sidetable.
8. Watches can be found in drawers and on dining tables. After finding the watches, place them on the desired shelf.
9. To clean a dish sponge, a plate, or a knife, use the sink basin. After cleaning the dish sponge, the plate, or the knife, place it on the shelf, countertop, or drawer respectively.
10. To cool a bowl or a mug, use the fridge. After cooling the bowl or the mug, place it in the cabinet.
11. Vases can be found on shelves. After finding the vase, take it to the desk where the desklamp is located and turn on the desklamp to examine the vase.
12. Eggs can be found in garbage cans. After finding the egg, heat it using the microwave before placing it back in the garbage can.
13. Apples can be found on countertops or in garbage cans. After finding the apple, cool it using the fridge before placing it in the microwave or heat it using the microwave before placing it back in the garbage can.
14. Forks can be found on countertops. After finding the fork, place it in the drawer.
15. Spoons can be found on countertops. After finding the spoon, clean it using the sink basin before placing it on the sidetable.
16. Lettuce can be found in the sink basin. After finding the lettuce, cool it using the fridge before placing it in the garbage can.
17. Bowls can be found in cabinets. After finding the bowl, cool it using the fridge before placing it in another cabinet.
18. Potatoes can be found on dining tables or in microwaves. After finding the potato, heat it using the microwave before placing it in the fridge or clean it using the sink basin before placing it on the sidetable.

19. Pillows can be found on beds. After finding the pillow, take it to the desk where the desklamp is located and turn on the desklamp to examine the pillow.
20. CDs can be found on dining tables. After finding the CD, place it on the sidetable.
21. Butterknives can be found on countertops or in drawers. After finding the butterknife, clean it using the sink basin before placing it on the countertop or dining table.
22. Credit cards can be found on sidetables. After finding the credit cards, place them on the dining table.
23. You can directly use a desklamp to examine an object without turning it on first.
24. Pots can be found in sink basins. After finding the pot, cool it using the fridge before placing it in the cabinet.

---

**ALFWorld: Human distilled tips and primitives**

Please consider the following tips to solve new tasks:

1. Try to execute a different action when you stuck in a loop of taking identical actions.
2. Try to search all other locations if the object could not be found in initial likely locations.
3. Remember the location of each object you have seen in order to quickly find it later.
4. Remember the number of objects you have seen in each location, so you can directly go back the location to find another one.
5. Carefully check the outcome of an action before taking next action.
6. You can directly put an object in/on an occupied receptacle.
7. You can only take actions to the object in hand or the objects in the current scene.
8. Do not mismatch target obejct with similar object, for example, a mug is different with a cup.
9. Do not express sorry/apology in your response.
10. Please only generate following actions in your response and strictly follow the syntax:

```
go to [receptacle] [id]
# Example:
    go to countertop 1
open [receptacle] [id]
# Example:
    open drawer 2
clean [object] [id] with [receptacle] [id]
# Example:
    clean lettuce 1 with sinkbasin 1
take [object] [id] from [receptacle] [id]
# Example:
    take lettuce 1 from countertop 1
close [receptacle] [id]
# Example:
    close drawer 2
heat [object] [id] with [receptacle] [id]
# Example:
    heat egg 1 with microwave 1
put [object] [id] in/on [receptacle] [id]
# Example:
    put lettuce 1 in/on countertop 1
use [object] [id]
# Example:
    use desklamp 1
cool [object] [id] with [receptacle] [id]
# Example:
    cool egg 1 with fridge 1
```

### E.5.3 Results in WebShop

---

**WebShop: Distilled policy improvement tips with contrastive prompt**

```
Please consider the following tips to solve new tasks:

1. You should strictly use the exact words of each clickable option without
making any changes.
2. Always ensure to select all the necessary options before making a purchase.
3. Pay attention to the specific requirements in the instruction, such as color
and size, and make sure to select them correctly.
4. Use the correct search terms to find the desired product. Avoid using
special characters like slashes (/) in the search term.
5. Make sure the product selected meets all the requirements in the
instruction.
6. Avoid making assumptions about the available options. Always select from the
options provided.
7. Always select the color or any other specific attribute mentioned in the
instruction before making a purchase.
8. Ensure to select the correct style or model of the product when multiple
options are available.
9. Check the product's features to ensure it meets all the requirements in the
instruction before making a purchase.
10. If the search results are not satisfactory, adjust the search terms instead
of repeating the same search.
11. Do not include the price limit in the search term. Instead, check the price
of each item in the search results.
12. If the first product selected does not meet all the requirements, go back
to the search results and select a different product.
13. Pay attention to the use of quotation marks in the search term to ensure
accurate results.
14. Only select the options that are specifically mentioned in the instruction.
Avoid selecting unnecessary options.
```

---

**WebShop: Distilled policy improvement tips with non-contrastive prompt**

```
Please consider the following tips to solve new tasks:

1. You should search with specific keywords, including the color or any
specific feature if required, select the relevant item to check and click
required attributes before placing the order.
2. Make sure to select the correct color and feature before purchasing the
item.
3. Always check the price of the item to ensure it is within the specified
budget.
4. If the first selected item does not meet the requirements, go back to the
search results and select another item. Repeat this process until a suitable
item is found.
5. Ensure to select the correct quantity or count of the item before
purchasing.
6. Make sure to select the correct size before purchasing the item.
7. When searching for a specific model of a product, include the model name in
the search keywords.
8. Ensure to select the correct flavor or variety when purchasing food items.
9. Check the material of the item if it is specified in the task.
10. If the item does not meet the requirements after checking its features, go
back and select another item.
11. When searching for a wig, include the color and material in the search
keywords.
12. Check the features of the product to ensure it meets the requirements
before purchasing.
13. Check the description of the product to ensure it meets the requirements
before purchasing.
```

```
14. Always check the product's features to ensure it meets the specific
requirements of the task.
15. After selecting an item, check its features to ensure it meets the task
requirements before purchasing.
```

**WebShop: Human distilled tips and primitives**

```
Please consider the following tips to solve new tasks:

1. Please pay attention to instruction description to find the product that
exactly matches with the requirement in the instruction;
2. You can click[Description] to find the detailed description of a product;
3. Remember to select all required attributes using click[Attributes] before
click[Buy Now];
4. You should click[Description] or click[Features] to double-check attributes
when there is no clickable attributes;
5. You should pay attention to the current observation to check clickable
buttons;
6. When the current selected product does not match the requirement in the
instruction well, you should navigate to the results page to check other likely
product candidates;
7. When there is no product that matches the requirement in the instruction
well, you should search again with a new query;
8. You should click[Back to Search] before re-searching with a new query;
9. Please only generate following actions in your response:

    search[Query]
    # Example: search[noise cancelling cosycost usb microphone]
    click[Attributes]
    # Example 1: click[16.9 fl oz (pack of 1)]
    # Example 2: click[blue]
    click[Product Title]
    # Example: click[B0972Q1T8T]
    click[Back to Search]
    # Example: click[Back to Search]
    click[< Prev]
    # Example: click[< Prev]
    click[Description]
    # Example: click[Description]
    click[Features]
    # Example: click[Features]
    click[Buy Now]
    # Example: click[Buy Now]
```

## E.6 Generated Code

### E.6.1 ALFWorld

Examples of generated code for each type of tasks under ALFWorld.

**ALFworld: Generated skill code**

```
def clean_object(specific_object_to_clean):
    # Go to the sinkbasin
    action = go('sinkbasin 1')
    env_observation = interact_with_environment(action)

    # Clean the object with the sinkbasin
    action = clean(specific_object_to_clean, 'sinkbasin 1')
    env_observation = interact_with_environment(action)
```

```python
def cool_object(specific_object_to_cool):
    # Go to the fridge
    action = go('fridge 1')
    env_observation = interact_with_environment(action)

    # Cool the object with the fridge
    action = cool(specific_object_to_cool, 'fridge 1')
    env_observation = interact_with_environment(action)

    return env_observation

def find_object(object_to_find, locations_to_search):
    for location in locations_to_search:
        action = go(location)
        env_observation, _ = interact_with_environment(action)
        if is_in(object_to_find, env_observation):
            specific_object_name = extract_pattern(env_observation,
                                                   object_to_find)
            return specific_object_name, location
        elif isclose(env_observation):
            action = open(location)
            env_observation, _ = interact_with_environment(action)
            if is_in(object_to_find, env_observation):
                specific_object_name = extract_pattern(env_observation,
                                                       object_to_find)
                return specific_object_name, location
    return None, None

def heat_object(specific_object_to_heat):
    # Go to the microwave
    action = go('microwave 1')
    env_observation = interact_with_environment(action)

    # Heat the specific object with the microwave
    action = heat(specific_object_to_heat, 'microwave 1')
    env_observation = interact_with_environment(action)

def put_object(specific_object_to_put, specific_target_location):
    # Go to the target location
    action = go(specific_target_location)
    env_observation, won = interact_with_environment(action)

    # If the target location is closed, open it
    if isclose(env_observation):
        action = open(specific_target_location)
        env_observation, won = interact_with_environment(action)

    # Put the object on the target location
    action = put(specific_object_to_put, specific_target_location)
    env_observation, won = interact_with_environment(action)

    # return whether env return won or not
    return won

def take_object(specific_object_to_take, pick_location):
    action = take(specific_object_to_take, pick_location)
    env_observation = interact_with_environment(action)
    return env_observation
```

```
def use_object(specific_object_to_use):
    action = use(specific_object_to_use)
    env_observation, won = interact_with_environment(action)
    return won
```

**ALFworld: Skill Composition**

```
# Based on the historical interaction and the new task description, the
sub-task sequence should be:
# find(bowl)
# take(bowl)
# clean(bowl)
# put(bowl, cabinet 1)

# Find the bowl
specific_object_name, location = find_object('bowl', ["cabinet 6", "cabinet
5", "cabinet 4", "cabinet 3", "cabinet 2", "cabinet 1", "coffeemachine 1",
"countertop 3", "countertop 2", "countertop 1", "drawer 3", "drawer 2", "drawer
1", "fridge 1", "garbagecan 1", "microwave 1", "shelf 3", "shelf 2", "shelf 1",
"sinkbasin 1", "stoveburner 4", "stoveburner 3", "stoveburner 2", "stoveburner
1", "toaster 1"])
# Take the bowl
take_object(specific_object_name, location)
# Clean the bowl
clean_object(specific_object_name)
# Put the bowl in cabinet 1
won = put_object(specific_object_name, 'cabinet 1')
```

### E.6.2   WebShop

In the WebShop, we have observed that the GPT-4 model excels in skill composition, effectively generating responses based on instructions and context. In contrast, the GPT-3.5 model exhibits proficiency in task decomposition, breaking tasks into multiple sub-tasks; however, it struggles to seamlessly integrate these skill functions within the broader context. For instance, it cannot effectively pass the output of one function as input to subsequent functions.

**Webshop: Generated skill code**

```
def select_item(items, instruction):
    # items: dict {'item number': discription}
    # instruction: str

    # Extract requirements from instruction
    requirements = extract_requirement(instruction)

    # Initialize a dictionary to store the number of satisfied requirements for
each item
    satisfied_requirements = {item: 0 for item in items.keys()}

    # Check each item against each requirement
    for item, description in items.items():
        for requirement in requirements:
            if requirement_match_check(description, requirement):
                satisfied_requirements[item] += 1

    # Select the item with the most satisfied requirements
    selected_item = max(satisfied_requirements, key=satisfied_requirements.get)
    return selected_item

def select_item_attributes(item_description, requirements):
```

```
    # item_description: string
    # requirements: [requirements#1,requirements#2,requirements#3,...]

    # Extract the most relevant attributes from the item description based on
the requirements
    relevant_attributes = select_most_relevant_attibutes(item_description,
requirements)

    # Return the list of relevant attributes
    return relevant_attributes
```

**Webshop: Skill Composition (GPT-4)**

```
# Sub-task sequence based on the historical interaction and new task
description:
# 1. search_item: search for a space-saving ottoman bench to match blue living
room
# 2. select_item: select the item that's 100x45x45cm and price lower than
410.00 dollars
# 3. select_item_attributes: select the attributes that match the requirements
# 4. purchase_item: buy the selected item

# Start the task
# Sub-task 1: search_item
instruction = "i'm looking for a space-saving ottoman bench to match my blue
living room. pick that one that's 100x45x45cm, and price lower than 410.00
dollars"
action = search(instruction)
env_obs, reward = interact_with_environment(action)

# Sub-task 2: select_item
items = parse_items_observation_into_items_dict(env_obs)
selected_item = select_item(items, instruction)
action = select(selected_item)
env_obs, reward = interact_with_environment(action)

# Sub-task 3: select_item_attributes
requirements = extract_requirement(instruction)
relevant_attributes = select_item_attributes(env_obs, requirements)
for attribute in relevant_attributes:
    action = click(attribute)
    env_obs, reward = interact_with_environment(action)

# Sub-task 4: purchase_item
action = buy()
env_obs, reward = interact_with_environment(action)

# Print the final reward
print(f"Final reward: {reward}")
```

**Webshop: Skill Composition (GPT-3.5)**

```
# Sub-task sequence:
# 1. search(instruction)
# 2. select_item()
# 3. select_item_attributes()
# 4. purchase_item()

# Code:
search('ultra hd motion detection surveillance dome camera color black size
:6mp wdr 2.8 mm, and price lower than 190.00 dollars')

select_item()
```

```
select_item_attributes(item_description, requirements)

purchase_item()
select_item()
```

