# OpenReview forum: "O3D: Offline Data-driven Discovery and Distillation for Sequential Decision-Making with Large Language Models"
_colmweb.org/COLM/2024/Conference — COLM_

### Official Review · Reviewer_Fifg · 2024-05-11

**Rating:** 7
**Confidence:** 4
**Ethics Flag:** 1

**Summary:**

This paper proposes an offline learning framework called O3D (Offline Data-driven Discovery and Distillation) that enables large language models (LLMs) to improve their performance on sequential decision-making tasks without fine-tuning.
The framework has three stages: a) Skill discovery by segmenting offline trajectories into sub-tasks. b) Knowledge distillation by discovering primitives and extracting policy improvement tips from contrasting successful and failed trajectories. c) Hierarchical policy execution where a base policy sequentially calls the learned skill-conditioned policies. Through iterative skill refinement and knowledge distillation, O3D allows LLMs to benefit from extensive and diverse offline data, improving their few-shot performance on challenging benchmarks like ALFWorld and WebShop. The proposed approach outperforms state-of-the-art baselines and demonstrates the effectiveness of utilizing offline data at scale to enhance LLM capabilities in sequential decision-making without fine-tuning.

**Questions To Authors:**

See weakness

**Reasons To Accept:**

1. O3D is a novel exploration aimed at enhancing the sequential decision-making performance of LLM agents through the utilization of large-scale offline data without the need for fine-tuning the model's weights.
2. It shows the capability to effectively identify and distill generalizable skills and valuable insights that can be applied to unseen downstream tasks.
3. It can also be utilized to augment these approaches with offline data. This possibility enables the achievement of high-quality policies prior to deployment while also reducing the cost of online interaction.
4. This paper is well-written and easy to follow.

**Reasons To Reject:**

1. The effectiveness of each stage is not clear to me. Further analysis of the design and experiments should be considered.
2. Following the previous question, how can the quality of the skill discovery be managed? LLMs can hallucinate.
3. What is the difference between O3D and other self-refine baselines like Reflexion?

---

> ### Author Rebuttal · Authors · 2024-05-31
>
> We greatly appreciate the valuable feedback and positive comments from Reviewer Fifg. Below we address reviewer's concerns and questions.
>
> > Q1. The effectiveness of each stage is not clear to me.
>
> Our Sec. 4.3 provides an ablation study to investigate the contribution of each component (see Fig. 4). Although the impact of each component is both task-dependent and model-dependent, the dash-dot line in Fig. 4 shows that our proposed components generally enhance performance across various tasks and models.
> We will further clarify these in the final version.
>
> > Q2. How can the quality of the skill discovery be managed? LLMs can hallucinate.
>
> We adopt several strategies to guarantee the quality of skill discovery: (1) provide offline trajectory data, ask LLM to first segment it into sub-trajectories, then label each sub-trajectory as a skill; (2) provide one demonstration in the prompt (see App. E.1.1) to guide LLM to discover the desired skills in a particular format; (3) iterate over multiple trajectories such that LLMs can refine the discovered skills, making sure the skills are general and non-duplicative.
>
> As the discovered skills are grounded on factual trajectories, we observe stable and high-quality skill discovery results without hallucination.
>
>
> > Q3. What is the difference between O3D and other self-refine baselines like Reflexion?
>
> Major differences between O3D and Reflexion are listed below:
>
> 1. Reflexion lets the LLM reflect purely based on online interaction history, and it takes multiple interaction trials to summarize tips. However, in practice, online interaction can be expensive or risky. In contrast, O3D learns from offline data. O3D achieves improvement in the first trial of deployment without extra online interactions.
> 2. Reflexion learns task-specific textual tips, only working for the current task. For example, the tip “in the next trial, I should put the apple on the countertop” cannot be directly transferred to a new task with a different goal, and such tips have to be re-generated for each downstream task. In contrast, O3D discovers reusable skills and generalizable textual tips. For example, the tip “I should strictly follow the task instruction and put the object in the desired location” can facilitate a wide range of downstream tasks.
> 3. Reflexion relies on online failure experiences, however, our approach leverages both success and failure experiences, which is more effective in generating proper tips to fix mistakes.

---

> > ### Comment · Reviewer_Fifg · 2024-06-01
> > **Thanks**
> >
> > Thanks the authors for the rebuttal. Most of my concerns have been resolved. I will maintain my positive score.

---

> > > ### Author Response · Authors · 2024-06-03
> > >
> > > Thank you very much for reading our response and giving positive feedback. We really appreciate it. We will polish the final version of the paper based on your constructive suggestions. Please let us know if there are any other questions. We are happy to discuss further.

---

### Official Review · Reviewer_JZjV · 2024-05-11

**Rating:** 7
**Confidence:** 4
**Ethics Flag:** 1

**Summary:**

This paper proposes an LLM system that can act in ALFWorld and WebShop interactive environments and improves on top of standard prompting baselines. The main idea stems for hierarchical modeling: by using prompted LLMs:
- **(1)** they first learn/segment subtasks from human demonstrations,
- **(2)** then automatically learn primitive actions (go to X, pick N X),
- **(3)** by using human demonstrations (has both positives and negatives) learn tips for the learned subtasks,
- **(4)** by using all of above, initial LLM decides what tasks to execute and another LLM executes each subtask.

The main technical contribution comes from automatic primitive skill learning and providing a way to use negative trajectories in prompting.

**Questions To Authors:**

1) Why not use "interactive decision making" instead of "sequential decision making" in the title and throughout the paper. I believe interactive part is more important than the sequential part?

2) Please add a small summary of the environment details to the main text (available actions, observations and states etc)

3) I quote: "This discrepancy may be attributed to the fact that GPT-3.5-0613 is fine-tuned with data and objectives that are diverged from the ALFWorld domain". This sentence seems to me very ambigious and generic. Aren't GPT-4 and GPT3.5 fine-tuned with same data?

**Reasons To Accept:**

1) Learning what abstractions (what kind of hierarchy) to use in sequential decision making is a hard problem, but the authors automatically discovers the abstractions for primitives and subtasks by using LLMs.

2) Their usage of both successfull and unsucessfull trajectories to genearate tips is an alternative to RL and clever in this context.

3) They improve over various baselines (ReACT, Reflexion) for this task.

4) Although the system complex, the paper does good job of ablating all the parts that I thought necessary in section 4.3

5) The writing of text is fluent.

**Reasons To Reject:**

1) What is the intuitive reason for worse performance in Table 3(b)
 comparing to ReACT-long? The authors could do more qualitative analysis here.

2) This is not a reason to reject but, in Figure 3, the discovered skills seems simple and easy to infer from data. It could be exciting to apply this method to more domains, see what are the skills that can be learned.

---

> ### Author Rebuttal · Authors · 2024-05-31
>
> We greatly appreciate the valuable feedback and positive comments from Reviewer JZjV. Below we address the reviewer’s concerns and questions.
>
> > Q1. What is the intuitive reason for worse performance in Table 3(b) comparing to ReACT-long?
>
> Table 3(b) shows that on WebShop, O3D achieves a higher success rate but a lower average score than ReAct-long. This is because the current implementation of O3D distills tips by comparing successful trajectories (score=1, when the bought item exactly matches the instruction) and failed trajectories (score<1, when the purchase failed or when the bought item does not exactly match the instruction). As a result, the distilled tips would guide the agent to achieve an exact match rather than maximizing the raw score.
>
> In contrast, ReAct-long utilizes the full trajectories with scores, it has a more fine-grained understanding of the relation between tasks and scores. One idea for improving the average score of O3D is to contrast the trajectories with higher scores and lower scores instead of simple successes and failures.
>
> > Q2. Not a reason to reject but it could be exciting to apply this method to more domains.
>
> Thank you for the suggestion. We also believe that the method can be applied to a broader range of domains, and we will investigate it in our future work. We will be excited to see follow-up work in various applications as well.
>
> > Q3. Why not use "interactive decision making" instead of "sequential decision making"?
>
> It’s an interesting point. Although the experiments are mainly conducted in interactive environments, the method can also be applied to non-interactive decision-making problems (using LLM), thus naming it as interactive decision-making would not be general enough.
>
>
> > Q4. Please add a small summary of the environment details to the main text.
>
> Thank you for the suggestion. We will make it more clear in the final version.
>
> > Q5. "This discrepancy may be attributed to the fact that GPT-3.5-0613 is fine-tuned with data and objectives that are diverged from the ALFWorld domain", which seems ambiguous and generic. Aren't GPT-4 and GPT3.5 fine-tuned with same data?
>
> We will clarify this in our final version. Here, we mean that GPT-3.5-0613 sometimes does not understand the objectives of ALFWorld tasks. For example, it sometimes wants to ask the house owner for help, which is not relevant to the ALFWorld domain. However, GPT-4 does a better job of understanding the restrictions of the environment.

---

> > ### Comment · Reviewer_JZjV · 2024-06-03
> > **thank you**
> >
> > Thank you for clarifying the metrics in Table 3(b), and your answers to my questions.
> >
> > I will maintain my positive score as is.

---

> > > ### Author Response · Authors · 2024-06-05
> > >
> > > Thank you very much for taking time reading our response and giving positive feedback. We are happy to discuss further if there are any other questions. We will polish the final version based on your constructive suggestions and comments.

---

### Official Review · Reviewer_KPqX · 2024-05-11

**Rating:** 7
**Confidence:** 3
**Ethics Flag:** 1

**Summary:**

The paper addresses the challenge of using Large Language Models (LLMs) for sequential decision-making tasks. Existing approaches often rely on high-quality few-shot demonstrations, but the limited context length of LLMs restricts their ability to process large-scale demonstrations with extended trajectories. To overcome this, the paper proposes a three-stage framework: First, it identifies reusable skills by segmenting offline datasets. Second, it enhances skill-conditioned policies through data distillation. The final stage involves constructing an interactive policy that utilizes the learned skills across various tasks. Experiments conducted on ALFWorld and WebShop demonstrate that this method surpasses traditional few-shot prompting baselines.

**Reasons To Accept:**

(1) The paper proposes an offline planning method, which differs from existing online methods.

(2) The proposed method not only achieves commendable empirical performance but also shows compatibility with existing prompting-based approaches, adding value to current methodologies in LLM-based decision-making.

(3) The paper is well-written.

**Reasons To Reject:**

(1) The paper comprises multiple components and stages, making it difficult to determine which component contributes the most.

(2) In Table 9 (Appendix D.2.1), the paper compares the proposed method with Demo2Code and AdaPlanner. The extremely low performance of AdaPlanner is unusual. I suggest adding results for AdaPlanner + GPT-4 to further validate the proposed method.

(3) Since the method uses offline data, it is important to study how much data O3D needs and how the quality of O3D affects performance.

---

> ### Author Rebuttal · Authors · 2024-05-31
>
> We greatly appreciate the valuable feedback and positive comments from Reviewer KPqX. Below we address the reviewer’s concerns and questions.
>
> > Q1：difficult to determine which component contributes the most.
>
> Sec. 4.3 shows an ablation study of each component. Although the impact of all components is task-dependent and model-dependent, dash-dot line in Fig. 4 shows that proposed components generally enhance performance across various tasks and models.
>
> > Q2：The extremely low performance of AdaPlanner is unusual.
>
> Thank you for the suggestion. Unfortunately, due to our budgetary constraint and the time limitation, we are unable to include AdaPlanner+GPT-4 at this point. However, we conducted additional experiments using GPT-3.5-0301,
>
> |Method|Pick|Clean|Heat|Cool|Look|Pick2|ALL|
> |---|---|---|---|---|---|---|---|
> |AdaPlanner|78|94|70|94|63|78|81|
> |O3D-code|96|84|91|86|89|88|89|
>
> AdaPlanner achieves reasonable performance as the original paper showed, while O3D-code can still outperform it by 8%.
>
> > Q3: how much data O3D needs and how the quality of O3D affects performance.
>
> 1. How much data: We discussed the data amount in App. A.1. We use a small batch size and iterate over multiple batches, and we can see the distilled tips converge only after a few iterations. So O3D does not need much data.
> 2. How the quality of O3D affects performance: In experiments, we observe reliable action discovery and tip distillation with the proposed trajectory contrasting method and iterative improvements. Even if the quality of Stage 1&2 is not optimal, the following points are noteworthy:
> - The primitive actions and tips are prepended to the normal prompt with few-shot examples used in ReAct. Since the distilled knowledge is from real trajectories, such additional context would be more helpful than harmful.
> - One can verify and modify the skills/actions/tips before deploying them to Stage 3, similar to verifying a trained ML model but the learned knowledge is in natural language and more interpretable. Compared to manual prompt design, O3D is more automatic and scalable.
> - We run additional experiments by inserting misleading information into the discovered and distilled results. The score of O3D (41%) does not drop with irrelevant information (49%), while slightly drops (38%) with misleading information, and is still much higher than baseline ReAct (7%) with GPT-3.5-0613. It shows the superior robustness of O3D. More details are in the response to Reviewer ubuS.

---

> > ### Comment · Reviewer_KPqX · 2024-06-04
> > **Thanks for additional experiments**
> >
> > I would like to thank the authors for conducting additional experiments comparing the proposed method with AdaPlanner using GPT-3.5-0301. I'm pleased to see that it demonstrates decent performance. As a result, I will adjust my score accordingly.

---

> > > ### Author Response · Authors · 2024-06-05
> > >
> > > Thank you very much for taking time reading our response and raising the score. We will polish the final version based on your constructive suggestions. Please let us know if there are any other questions. We are happy to discuss further.

---

### Official Review · Reviewer_ubuS · 2024-05-13

**Rating:** 5
**Confidence:** 4
**Ethics Flag:** 1

**Summary:**

This paper proposed a prompting framework that can 1) automatically discover subtasks/skills along with their primitives and preconditions from trajectories, 2) apply discovered skills to new tasks in a hierarchical way. More specifically, for prediction, LM will first figure out the subtasks/skills and then retrieve relevant examples from the constructed database on how to use primitive actions to complete a task. Results based on GPT models on ALFWorld and WebShop demonstrate that it can improve upon ReAct.

Though the general idea and presented results are interesting, the practical value of this work is unclear due to the limited experiment setting (see below).

**Questions To Authors:**

For reproducibility, it would be better to also provide results on one open-source models.

For the baseline ReAct-Long, I wonder the performance of its retrieval-based variant --retrieving relevant trajectories based on task descriptions. It's well-known that retrieval is crucial for few-shot in-context learning methods.

**Reasons To Accept:**

The general idea of discovering skills based on trajectory data is interesting, and this paper reveals the potential of using offline data to discover better system prompt (i.e., with macro/template actions) for embodied tasks.

**Reasons To Reject:**

The main components DiscoverPrompt and DistillPrompt all rely on LMs, and there seems no automatic verification to check how good are the distilled skills and so-called tips.  I'm concerned about 1) how often LMs fail to find generalisable actions (e.g., not too specific actions) or even wrong actions, 2) what if DistillPrompt provides wrong preconditions/corrections of discovered actions. Since these skills will serve as system prompt for new tasks, the errors in them will potentially propagate to all predictions.  Overall, I think how to ensure the quality of discovered skills is an important question, but there seems no experiments/discussion on it.

---

> ### Author Rebuttal · Authors · 2024-05-31
>
> We greatly appreciate the valuable feedback from Reviewer ubuS, and address the concerns below.
>
> > Q1: how often LMs fail to find generalisable actions or wrong actions.
>
> In experiments, we find GPT4 is very reliable in discovery and distillation, with a few strategies we used:
> 1. Provide few-shot examples (App. E.1.2), so the model can better understand the goal.
> 2. Iteratively run discovery/distillation to keep refining summarized skills/actions/tips, so potential mistakes can be corrected in later iterations.
> 3. Trajectory contrasting (compares success and failure trails), so the summarized knowledge is grounded on facts rather than hallucination.
>
> > Q2: what if DistillPrompt provides wrong preconditions/corrections of discovered actions
>
> To test O3D’s robustness to errors, we run additional experiments by inserting misleading information into the discovered and distilled results.
> - Setting 1: Irrelevant Information. For every skill, we insert an action and a tip from a different skill, which is not useful to the current skill and may be distractors.
> - Setting 2: Misleading Information. For every skill, we insert a wrong action and an irrelevant tip that are not even valid in ALFWorld.
>
> Results: we run GPT-3.5-0613 on ALFWorld, where the scores are:
> - Original O3D: 41%
> - O3D + Irrelevant Info: 49%
> - O3D + Misleading Info: 38%
>
> We can see that with the irrelevant or misleading info, the agent can still achieve high results (ReAct is 7%). Note that our distilled knowledge is summarized from real trajectories and provides additional context to the basic few-shot examples, so it can be more helpful than harmful. Moreover, Stage 1&2 of O3D are offline, so one can check, verify and modify the learned skills/actions/tips (on held-out validation sets) before deploying them to Stage 3 interactions.
>
> > Q3: It would be better to also provide results on one open-source model.
>
> Thank you for the suggestion. We will release the codebase and add open-source models in the final version.
>
> > Q4: The performance of ReAct-Long’s retrieval-based variant.
>
> This is an interesing point.
> 1. ReAct uses manually selected relevant examples as prompts, which is already a retrieval-based baseline method.
> 2. It is a great future direction to explore more fine-grained retrieval-based methods for long-horizon interactive tasks, but it is non-trivial and out of our current scope.
> 3. O3D can be combined with retrieval methods, similar to how O3D+Reflexion improves Reflexion (Table 2).

---

> > ### Comment · Reviewer_ubuS · 2024-05-31
> >
> > Thanks for the rebuttal.  I've increased the score given the new results and promised experiments.

---

> > > ### Author Response · Authors · 2024-06-03
> > >
> > > Thank you very much for reading our response and raising the score. We are keeping running the suggested experiments and will add them to the final version of the paper.
> > >
> > > We also finished another robustness test on WebShop. Similarly to the previous experiments on ALFWorld, we added irrelevant information and misleading information to the distilled knowledge. For example, we insert the action “rate [Item]” and tip “You can rate the items and leave feedback” which are not valid in the environment and can be misleading. We add these distractors to every skill-conditioned policy, so that it is a strong perturbation to the original method. In this way, we can simulate the case where the LLM hallucinated in the discovery and distillation process. The resulted success rates with GPT-3.5-0613 are as below.
> > >
> > > - Original O3D: 35%
> > > - O3D + Irrelevant Info: 31%
> > > - O3D + Misleading Info: 34%
> > >
> > > Compared to the baseline ReAct whose success rate is 27%, our O3D still achieves higher performance even when distractors exist. There suggests that **Stage 3 of O3D is robust to noise or mistakes from the first 2 stages, if any.**
> > >
> > > In addition, we also would like to emphasize that **our algorithm design makes the first 2 stages grounded on facts in offline trajectories, so that the skills, actions and tips are relatively reliable.** Such design includes iterative discovery and distillation, trajectory contrasting, as well as segmentation-based skill summarization. For example, instead of asking the LLM to think what is the mistake and how to fix it (a common reflective method) which may lead to hallucination, we ask the LLM to compare a success sub-trajectory and a failure sub-trajectory and find their differences. In this way, it is much easier for the LLM to locate the key to success in the environment.
> > >
> > > We hope the above experiment results and explanations have addressed the reviewer’s concern on the quality control and robustness of the method. Please let us know if there is any remaining question or concern. We are very happy to discuss further. Thank you again for your time and suggestions.

---

### Comment · Area_Chair_qej6 · 2024-06-03
**Discussion period**

Hi everyone! Thanks to reviewers ubuS and Fifg for your reply to author response. For the other reviewers, I'd appreciate it if you could look through the authors' comments soon and let us know if your concerns have been addressed / if there are any outstanding issues soon so there's time to discuss.

With appreciation,
Your AC

---

### Decision · Program_Chairs · 2024-07-10

**Decision:**

Accept

**Comment:**

This paper describes a method for discovering reusable skills from offline demonstrations, using an LLM to parse and segment demonstrations, then distilling each segmented-out skill into a dedicated policy. Reviewers agree that the proposed method is interesting and complementary to current work on deliberation methods for "LLM agents", and is convincingly evaluated. I think it will be a great fit for the conference.

One thought I had while reading the paper & reviews was that it might be nice to also include some discussion of pre-giant-LM methods that used language as a scaffold for skill learning, e.g. https://arxiv.org/abs/2110.07342, https://arxiv.org/abs/2110.01517, https://arxiv.org/abs/1903.01973.